# The route to chaos in routing games:
# When is price of anarchy too optimistic?

**Thiparat Chotibut**
Chula Intelligent and Complex Systems
Department of Physics, Faculty of Science
Chulalongkorn University
`Thiparat.C@chula.ac.th`

**Fryderyk Falniowski**
Department of Mathematics
Cracow University of Economics
`falniowf@uek.krakow.pl`

**Michał Misiurewicz**
Department of Mathematical Sciences
Indiana Univ.-Purdue Univ. Indianapolis
`mmisiure@math.iupui.edu`

**Georgios Piliouras**
Engineering Systems and Design
Singapore Univ. of Technology and Design
`georgios@sutd.edu.sg`

## Abstract

Routing games are amongst the most studied classes of games in game theory. Their most well-known property is that learning dynamics typically converge to equilibria implying approximately optimal performance (low Price of Anarchy). We perform a stress test for these classic results by studying the ubiquitous learning dynamics, Multiplicative Weights Update (MWU), in different classes of congestion games, uncovering intricate non-equilibrium phenomena. We study MWU using the actual game costs without applying cost normalization to $[0, 1]$. Although this non-standard assumption leads to large regret, it captures realistic agents' behaviors. Namely, as the total demand increases, agents respond more aggressively to unbearably large costs.

We start with the illustrative case of non-atomic routing games with two paths of linear cost, and show that every system has a carrying capacity, above which it becomes unstable. If the equilibrium flow is a symmetric $50 - 50\%$ split, the system exhibits one period-doubling bifurcation. Although the Price of Anarchy is equal to one, in the large population limit the time-average social cost for all but a zero measure set of initial conditions converges to its worst possible value. For asymmetric equilibrium flows, increasing the demand eventually forces the system into Li-Yorke chaos with positive topological entropy and periodic orbits of all possible periods. Remarkably, in all non-equilibrating regimes, the time-average flows on the paths converge *exactly* to the equilibrium flows, a property akin to no-regret learning in zero-sum games. We extend our results to games with arbitrarily many strategies, polynomial cost functions, non-atomic as well as atomic routing games, and heterogenous users.

## 1  Introduction

Congestion and routing games [44] are amongst the most well studied class of games in game theory. Being isomorphic to potential games [41], congestion games are one of the few classes of games in which a variety of learning dynamics are known to converge to Nash equilibria [5, 19, 22, 23, 32].

Congestion games also play a pivotal role in the study of Price of Anarchy [33, 46]. Price of Anarchy (PoA) is defined as the ratio of the social cost of the worst Nash equilibrium to the optimal social cost. A small Price of Anarchy implies that all Nash equilibria are near optimal, and hence any

equilibrating learning dynamics suffices to reach approximately optimal system performance. One of the hallmarks of the PoA research has been the development of tight PoA bounds for congestion games that are independent of the topology of the network or the number of users. Specifically, under the prototypical assumption of linear cost functions, Price of Anarchy in the case of non-atomic agents (in which each agent controls an infinitesimal amount of flow) is at most $4/3$ [46]. In the atomic case (in which each agent controls a discrete unit of flow), it is at most $5/2$ [15], with small networks sufficing to provide tight lower bounds.

Additionally, congestion games have paved the way for recent developments in Price of Anarchy research, extending our understanding of system performance even for non-equilibrating dynamics. Roughgarden [45] showed that most Price of Anarchy results could be organized in a common framework known as $(\lambda, \mu)$-smoothness. For classes of games that satisfy this property, such as congestion games, the Price of Anarchy bounds derived for worst case Nash equilibria immediately carry over to worst case instantiations of regret minimizing algorithms. Several online learning algorithms are known to belong to this class including the well known Multiplicative Weights Update (MWU) [3, 24]. This seems to suggest that MWU behavior in congestion games is more or less understood, always guaranteeing approximate optimality.

MWU, however, only offers such guarantees under a specific set of assumptions: i) the cost range $C$ is normalized to lie in $[0, 1]$ ii) the base of the exponential update step $(1 - \epsilon) = e^{-a} \approx 1$, or equivalently, both $\epsilon$ and $a$ are close to zero, or, ideally, decreasing with time. These assumptions, although standard in the online optimization literature, are far from the norm in behavioral economics [12]. For example, arguably one of the most well known learning models in behavioral game theory is Experienced Weighted Attraction (EWA) [12, 29–31]. This model includes as a special case the MWU algorithm (see Supplementary Material (SM) C for more discussion on modelling). In EWA, when the payoff sensitivity parameter $a$ is inferred from experimental data, it can vary widely from game to game, and in many cases $a \gg 1$.[1] In this range of $a$'s (resp. $\epsilon$) the black-box MWU regret bounds do not apply. Although these parameter ranges are evidently interesting from a behavioral game theory perspective, very little is known about them.

In the case of congestion games, there exist even more pressing reasons to study MWU without enforcing any normalizing assumption on $C$ or $\epsilon$.[2] Studying these models are necessary if we are to fully explore the effects of increased demand on system stability and efficiency. As the total traffic in a road network increases, drivers experience increased delays. As they do, their behaviors should reasonably change; e.g. doubling the total demand in a linear network doubles its range of costs, and its daily experienced costs, which should result in more aggressive behavioral responses from the increasingly agitated agents. Even though several recent papers study the effects of increased demand/population size on PoA in congestion games [16, 17, 21] and show improved performance (lower PoA) in the large population limit, *no systematic study of the effects of increased total demand on learning* has been performed. This is partly because in its full generality this question lies outside the machinery of standard regret minimization techniques.

As a first step to understand the effects of increased demand, let's examine why it leads to lower PoA. Let us consider the simplest example; a congestion game with two strategies and two agents where the cost/latency of each strategy is equal to its load. The worst Nash equilibrium of this game has both agents choosing a strategy uniformly at random. The expected cost of each agent is $3/2$, i.e., a cost equal to 1 due to their own load, and an expected extra cost of $1/2$ due to the 50% chance of adopting the same strategy as the other agent. On the other hand, at the optimal state, each agent selects a distinct strategy at a cost of 1. As a result, the PoA for this game is $3/2$. Suppose now that we increase the number of agents from 2 to $N$.[3] The worst equilibrium still has each agent choosing a strategy uniformly at random at an expected cost of $(N - 1)/2 + 1 = (N + 1)/2$. The optimal configuration splits the agents, deterministically and equally to both strategies at a cost of $N/2$ per

agent. The PoA is $1 + 1/N$, converging to 1 as $N$ grows. As the population size grows, the atomic game is more conveniently described by its effective non-atomic counterpart, with a continuum of users, and a unique equilibrium that equidistributes the total demand $N$ between the two strategies. So, the equilibria indeed are effectively optimal. How does this large demand, however, affect the dynamics?

**Informal Meta-Theorem:** We analyze MWU in routing/congestion games under a wide range of settings and combinations thereof (two/many paths, non-atomic/atomic, linear/polynomial, etc). Given any such game $G$ and an arbitrary learning rate (step-size) $\epsilon \in (0, 1)$, we show that there exist a system capacity $N_0(G, \epsilon)$ such that *if the total demand exceeds this threshold the system is provably unstable* with complex non-equilibrating behavior. The existence of both *periodic behavior* as well as *chaotic behavior* is proven and we give formal guarantees about the conditions under which they emerge. Despite this unpredictability of the non-equilibrating regimes, the *time-average* flows exhibit regularity and for linear costs *converge to equilibrium*. The variance, however, of the resulting non-equilibrium flows leads to increased inefficiencies, as the *time-average cost can be arbitrarily high*, even for simple games where all equilibria are optimal.

**Intuition behind instability:** To build an intuition of why instability can arise, let us revisit the simple example with two strategies and let us consider a continuum/large number of users updating their strategies according to a learning dynamic, e.g. MWU with a step-size $\epsilon$. Given any non-equilibrium initial condition, the agents on the over-congested strategy have a strong incentive to migrate to the other strategy. As they all act in unison, *if the total demand is sufficiently large*, the corrective deviation to the other strategy will be overly aggressive, resulting in the other strategy becoming over-congested. With this heuristic consideration, a self-sustaining non-equilibrating behavior, where users bear higher time-average costs than those at the equilibrium flow, is indeed plausible. In this work, we show that it is in fact provably true, even for games with arbitrarily many strategies. In addition to the proof of instability, we provide detailed theoretical and numerical investigation of the emergent dynamics and the onset of chaos, and introduce a host of dynamical systems techniques in the study of games (see Supplementary Materials (SM) A for dynamical systems background).

**Base model & results:** We start by focusing on the minimal case of linear non-atomic congestion games with two edges and total demand $N$. All agents are assumed to evolve their behavior using Multiplicative Weights Updates with an arbitrarily small, fixed learning rate $\epsilon$. In Section 3 we prove that every such system has a critical threshold, a hidden system capacity, which when exceeded, the system exhibits a bifurcation and no longer converges to its equilibrium flow. If the unique equilibrium flow is the $50-50\%$ split (doubly symmetric game), the system proceeds through exactly one period-doubling bifurcation, where a single attracting periodic orbit of period two replaces the attracting fixed point. In the case where the game possesses an asymmetric equilibrium flow, the bifurcation diagram is much more complex (Figure 1 in the main text and Figure 2 in SM B). As the total demand changes, we will see the birth and death of periodic attractors of various periods. All such systems provably exhibit Li-Yorke chaos, given sufficiently large total demand. This implies that there exists an uncountable set of initial conditions such that the set is "scrambled", i.e., given any two initial conditions $x(0), y(0)$ in this set, $\liminf dist(x(t), y(t)) = 0$ while $\limsup dist(x(t), y(t)) > 0$.

Everywhere in the non-equilibrating regime, MWU's time-average behavior is reminiscent of its behavior in zero-sum games. Namely, the time-average flows and costs of the strategies converge *exactly* to their *equilibrium values*. Unlike zero-sum games, however, these non-equilibrium dynamics exhibit large regret (Section 4), and (possibly arbitrarily) high time-average social costs (Section 5), even when the Price of Anarchy is equal to one. In SM A, we argue that the system displays another signature of chaotic behavior, positive topological entropy. We provide an intuitive explanation by showing that if we encode three events: A) the system is approximately at equilibrium, B) the first strategy is overly congested, C) the second strategy is overly congested, then the number of possible sequences on the alphabet $\{A, B, C\}$, encapsulating possible system dynamics, grows exponentially with time. Clearly, if the system reached an (approximate) equilibrium, any sequences must terminate with an infinite string of the form $\dots AAA\dots$. Instead, we find that the system can become truly unpredictable. In SM F, we show that the system may possess multiple distinct attractors and hence the time-average regret and social cost depend critically on initial conditions. Properties of periodic orbits, the evidence of Feigenbaum's universal route to chaos in our non-unimodal map, are also provided.

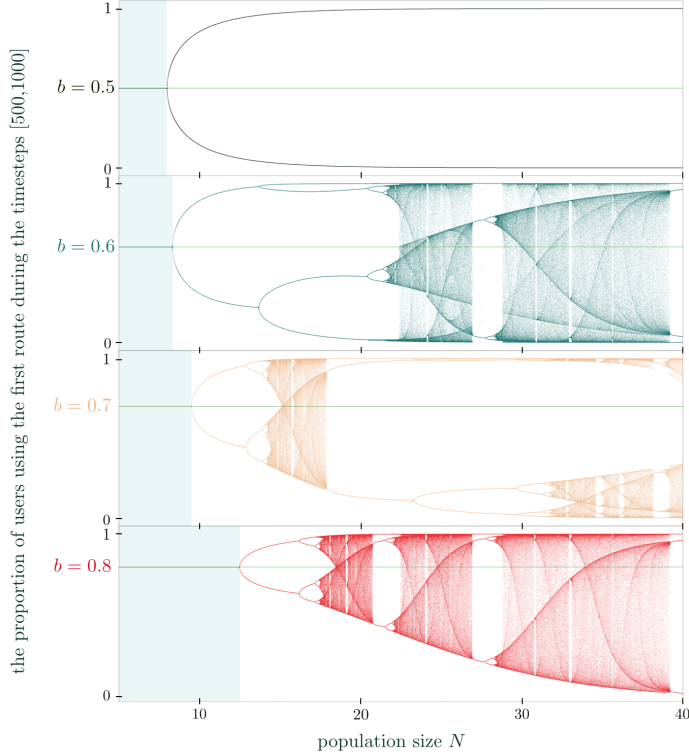

Figure 1: Bifurcation diagrams summarizing the non-equilibrium phenomena identified in this work. When Multiplicative Weights Update (MWU) learning is applied on a non-atomic linear congestion games with two routes, population increase drives period-doubling instability and chaos. Standard equilibrium analysis only holds at small population sizes, shown in light cyan regions. As population size $N$ (up to a rescaling factor of a fixed learning rate) increases, regret-minimizing MWU algorithm *no longer converges* to the Nash equilibrium flow $b$, depicted as the green horizontal lines; the proportion of users using the first route deviates significantly from the Nash equilibrium flow. When the equilibrium flow is symmetric between the two routes ($b = 0.5$), large $N$ leads to non-equilibrium dynamics that is attracted toward a limit cycle of period two. For large $N$, the two periodic points approach 1 or 0 arbitrarily close, meaning that almost all users will occupy the same route, while simultaneously alternating between the two routes. Thus, the time-average social cost can become as bad as possible. In *any* game with an asymmetric equilibrium flow ($b \neq 0.5$), Li-Yorke chaos is inevitable as $N$ increases. Although the dynamics is non-equilibrating and chaotic, the time-average of the orbits still converges *exactly* to the equilibrium $b$. This work proves the aforementioned statements, and investigates the implications of non-equilibrium dynamics on the standard Price of Anarchy analysis.

**Extensions:** In SM G, we prove that our results hold not only for graphs with two paths but extend for arbitrary number of paths. In SM H, we prove Li-Yorke chaos, positive topological entropy and time-average results for polynomial costs. In SM I, we provide extensions for heterogeneous users. Finally, in SM J, we produce a reduction for MWU dynamics from atomic to non-atomic congestion games. This allows us to extend our proofs of chaos, inefficiency to atomic congestion games with many agents.

## 2 Model

We consider a two-strategy *congestion game* (see [44]) with a continuum of players (agents), where all of them apply the *multiplicative weights update* to update their strategies [3]. Each of the players controls an infinitesimal small fraction of the flow. The total flow of all the players is equal to $N$. We will denote the fraction of the players adopting the first strategy at time $n$ as $x_n$.

**Linear routing games**: The cost of each path (link, route, or strategy) here will be assumed proportional to the *load*. By denoting $c(j)$ the cost of selecting the strategy number $j$ (when $x$ fraction of the agents choose the first strategy), if the coefficients of proportionality are $\alpha, \beta > 0$, we obtain

$$c(1) = \alpha N x, \qquad\qquad c(2) = \beta N (1 - x). \tag{1}$$

Without loss of generality we will assume throughout the paper that $\alpha + \beta = 1$. Therefore, the values of $\alpha$ and $\beta = 1 - \alpha$ indicate how different the path costs are from each other. Our analysis on the emergence of bifurcations, limit cycles and chaos will carry over immediately to the cost functions of the form $\alpha x + \gamma$. As we will see, the only parameter that is important is the value of the equilibrium split, i.e. the percentage of players using the first strategy at equilibrium. The first advantage of this formulation is that the fraction of agents using each strategy at equilibrium is independent of the flow $N$. The second advantage is that the Price of Anarchy of these games is exactly 1, independent of $\alpha, \beta$, and $N$. Hence, our model offers a natural benchmark for comparing equilibrium analysis, which suggests optimal social cost, to the time-average social cost arising from non-equilibrium learning dynamics which as we show can be as large as possible.

## 2.1 Learning in congestion games with multiplicative weights

At time $n + 1$, we assume the players know the cost of the strategies at time $n$ (equivalently, the probabilities $x_n, 1 - x_n$) and update their choices. Since we have a continuum of agents, the realized flow (split) is accurately described by the probabilities $(x_n, 1 - x_n)$. The algorithm for updating the probabilities that we focus on is the *multiplicative weights update* (MWU) the ubiquitous learning algorithm widely employed in Machine Learning, optimization, and game theory (also known as Normalized Exponentiated Gradient) [3, 50].

$$x_{n+1} = \frac{x_n(1-\epsilon)^{c(1)}}{x_n(1-\epsilon)^{c(1)} + (1-x_n)(1-\epsilon)^{c(2)}} = \frac{x_n}{x_n + (1-x_n)(1-\epsilon)^{c(2)-c(1)}}. \tag{2}$$

In this way, a large cost at time $n$ will decrease the probability of choosing the same strategy at time $n + 1$. By substituting into (2) the values of the cost functions from (1) we get:

$$x_{n+1} = \frac{x_n(1-\epsilon)^{\alpha N x_n}}{x_n(1-\epsilon)^{\alpha N x_n} + (1-x_n)(1-\epsilon)^{\beta N(1-x_n)}} = \frac{x_n}{x_n + (1-x_n)(1-\epsilon)^{\beta N - N x_n}}.$$

We introduce the new variables

$$a = N \ln\left(\frac{1}{1-\epsilon}\right), \quad b = \beta. \tag{3}$$

We will thus study the dynamical systems generated by the one-dimensional map:

$$f_{a,b}(x) = \frac{x}{x + (1-x)\exp(a(x-b))}. \tag{4}$$

The learning rate $\epsilon$ in MWU can be regarded as a fixed constant in the following analysis but its exact value is not of particular interest as our analysis/results will hold for any fixed choice of $\epsilon$ no matter how small/large. Setting $\epsilon = 1 - 1/e$ such that $\ln(1/(1-\epsilon)) = 1$ simplifies notation as under this assumption $a = N$. We will then study the effects of the remaining two parameters on system performance, i.e. $a$, the *(normalized) system demand* and $b$, the *(normalized) equilibrium flow*. When $b = 0.5$ the routing game is fully symmetric; whereas, when $b$ is close to 0 or 1, the routing instance becomes close to a Pigou network with almost all agents selecting the same edge at equilibrium.

## 2.2 Regret, Price of Anarchy and time average social cost

**Regret**: Fix cost vectors $\mathbf{c}_1, \ldots, \mathbf{c}_T$. The (expected) *regret* of the (randomized) algorithm choosing actions according $x_1, \ldots, x_T$ is

$$\underbrace{\sum_{n=1}^{T} \mathbf{E}_{a_n \sim x_n} c_n(a_n)}_{\text{our algorithm}} - \underbrace{\min_{a \in A} \sum_{n=1}^{T} c_n(a)}_{\text{best fixed action}}, \tag{5}$$

where $\mathbf{E}_{a_n \sim x_n} c_n(a_n)$ expresses the expected cost of the algorithm in time period $n$, when an action $a_n \in A$ is chosen according to the probability distribution $x_n$. In our setting, Eq. (5) translates to:

$$\sum_{n=1}^{T} \left( \alpha N x_n^2 + \beta N (1 - x_n)^2 \right) - \min \left\{ \sum_{n=1}^{T} \alpha N x_n, \sum_{n=1}^{T} \beta N (1 - x_n) \right\}.$$

**Price of Anarchy:** The *Price of Anarchy* of a game is the ratio of the supremum of the social cost over all Nash equilibria divided by the social cost of the optimal state, where the social cost of a state is the sum of the costs of all agents. In our case, if a fraction $x$ of the population adopts the first strategy, the social cost is $SC(x) = \alpha N^2 x^2 + \beta N^2 (1 - x)^2$. In non-atomic congestion games, it is well known that all equilibria have the same social cost. Moreover, for linear cost functions $c_1(x) = \alpha x$, $c_2(x) = \beta x$ the PoA is equal to one, as the unique equilibrium flow, $\beta$, is also the unique minimizer of the social cost, which attains the value.

Since MWU is not run with a decreasing step-size, the time-average regret may not vanish, so standard techniques implying equilibration do not apply [9] and a more careful analysis is needed. As we will show, as the total demand increases, the system will bifurcate away from the Nash equilibrium; and the time-average social cost will be strictly greater than its optimal value (in fact it can be arbitrarily close to its worst possible value). Taking a dynamical systems point of view, we will also study typical dynamical trajectories, since simulations suffice to identify the limits of time-averages of these trajectories, which occur for initial conditions with positive Lebesgue measure. We define the normalized time-average social cost as follows:

$$\frac{\text{Time-average social cost}}{\text{Optimum social cost}} = \frac{\frac{1}{T} \sum_{n=1}^{T} \left( \alpha N^2 x_n^2 + \beta N^2 (1 - x_n)^2 \right)}{N^2 \alpha \beta} = \frac{\frac{1}{T} \sum_{n=1}^{T} \left( x_n^2 - 2\beta x_n + \beta \right)}{\beta (1 - \beta)}. \tag{6}$$

# 3 Limit cycles and chaos, with time-average convergence to Nash equilibrium

This section discusses the behavior of the one-dimensional map defined by (4), and its remarkable time-average properties, which we will later employ to analyze the time-average regret and the normalized time-average social cost in Sections 4 and 5. The map generated by non-atomic congestion games here reduces to the map studied in [14], in which two-agent linear congestion games are studied. Up to redefinition of the parameters as well as with the symmetric initial conditions, i.e., on the diagonal, the one-dimensional map in the two scenarios are identical. Thus in this section, we restate some key properties of the map. For the proofs we refer the reader to [14].

We will start by investigating the dynamics under the map (4) with $a > 0$, $b \in (0, 1)$. It has three fixed points: $0$, $b$ and $1$ (see the middle column of Figure 2 in SM B). The derivatives at the three fixed points are

$$f_{a,b}'(0) = \exp(ab), \quad f_{a,b}'(1) = \exp(a(1 - b)), \quad f_{a,b}'(b) = ab^2 - ab + 1.$$

Hence, the fixed points $0$ and $1$ are repelling, while $b$ is repelling whenever $a > 2/b(1 - b)$ and attracting otherwise.

The critical points of $f_{a,b}$ are solutions to $ax^2 - ax + 1 = 0$. Thus, if $0 < a \leq 4$, then $f_{a,b}$ is strictly increasing. If $a > 4$, it has two critical points

$$x_l = \frac{1}{2} \left( 1 - \sqrt{1 - \frac{4}{a}} \right), \quad x_r = 1 - x_l = \frac{1}{2} \left( 1 + \sqrt{1 - \frac{4}{a}} \right) \tag{7}$$

so $f_{a,b}$ is bimodal.

Let us investigate regularity of $f_{a,b}$. Nice properties of interval maps are guaranteed by the negative Schwarzian derivative. Let us recall that the Schwarzian derivative of $f$ is given by the formula

$$Sf = \frac{f'''}{f'} - \frac{3}{2} \left( \frac{f''}{f'} \right)^2.$$

A "metatheorem" states that almost all natural noninvertible interval maps have negative Schwarzian derivative. Note that if $a \leq 4$ then $f_{a,b}$ is a homeomorphism, so we should not expect negative Schwarzian derivative for that case.

**Proposition 3.1.** *If $a > 4$ then the map $f_{a,b}$ has negative Schwarzian derivative.*

For maps with negative Schwarzian derivative each attracting or neutral periodic orbit has a critical point in its immediate basin of attraction. Thus, we know that if $a > 4$ then $f_{a,b}$ can have at most two attracting or neutral periodic orbits.

## 3.1 Time-average convergence to Nash equilibrium $b$

While we know that the fixed point $b$ is often repelling, especially for large values of $a$, we can show that it is attracting in a time-average sense.

**Definition 3.2.** *For an interval map $f$ a point $p$ is* Cesàro attracting *if there is a neighborhood $U$ of $p$ such that for every $x \in U$ the averages $\frac{1}{T} \sum_{n=0}^{T-1} f^n(x)$ converge to $p$.*

We can show that $b$ is globally Cesàro attracting. Here by "globally" we mean that the set $U$ from the definition is the interval $(0, 1)$.

**Theorem 3.3.** *For every $a > 0$, $b \in (0, 1)$ and $x \in (0, 1)$ we have $\lim_{T \to \infty} \frac{1}{T} \sum_{n=0}^{T-1} f_{a,b}^n(x) = b$.*

**Corollary 3.4.** *For every periodic orbit $\{x_0, x_1, \ldots, x_{T-1}\}$ of $f_{a,b}$ in $(0, 1)$ its center of mass (time average) $\frac{x_0 + x_1 + \cdots + x_{T-1}}{T}$ is equal to $b$.*

Applying the Birkhoff Ergodic Theorem (see SM A), we get a stronger corollary.

**Corollary 3.5.** *For every probability measure $\mu$, invariant for $f_{a,b}$ and such that $\mu(\{0, 1\}) = 0$, we have $\int_{[0,1]} x \, d\mu = b$.*

These statements show that the time average of the orbits generated by $f_{a,b}$ converges exactly to the Nash equilibrium $b$, no matter what the initial point is. Next, we discuss what happens as we fix $b$ and increase the total demand by letting $a$ grow large.

## 3.2 Periodic orbits and chaotic behavior

Now we focus on the behavior of $f_{a,b}$-trajectories of points from $(0, 1)$.

**Theorem 3.6.** *For $a \leq 2/b(1-b)$ trajectories of all points of $(0, 1)$ converge to Nash equilibrium $b$.*

As nothing interesting is happening for small values of $a$, we turn our interest to large $a$.

When $b = 0.5$, the coefficients of the cost functions are identical, i.e., $\alpha = \beta$. See Figure 1 and Figures 2 and 4 in SM B.

**Theorem 3.7.** *If $a > 2/b(1-b)$ then $f_{a,0.5}$ has a periodic attracting orbit $\{\sigma_a, 1 - \sigma_a\}$, where $0 < \sigma_a < 0.5$. This orbit attracts trajectories of all points of $(0, 1)$, except countably many points whose trajectories eventually fall into the repelling fixed point $0.5$.*

Theorem 3.7 together with Theorem 3.6 state that every trajectory converges to the Nash equilibrium $b = 0.5$ as long as $b$ is attracting. At the moment when $b$ becomes repelling ($a > \frac{2}{b(1-b)} = 8$), a periodic orbit of period 2 attracting almost all points is created and no longer trajectories converge to the Nash equilibrium.

Now we proceed with the case when $b \neq 0.5$, that is, when the cost functions differ. See Figure 1, as well as Figures 2, 3 in SM B. We fix $b \in (0, 1) \setminus \{0.5\}$ and let $a$ go to infinity. We will show that if $a$ becomes sufficiently large (but how large, depends on $b$), then $f_{a,b}$ is Li-Yorke chaotic and has periodic orbits of all possible periods.

**Definition 3.8** (Li-Yorke chaos). *Let $(X, f)$ be a dynamical system and $x, y \in X$. We say that $(x, y)$ is a* Li-Yorke pair *if*

$$\liminf_{n \to \infty} dist(f^n(x), f^n(y)) = 0, \text{ and } \limsup_{n \to \infty} dist(f^n(x), f^n(y)) > 0.$$

*A dynamical system $(X, f)$ is* Li-Yorke chaotic *if there is an uncountable set $S \subset X$ (called* scrambled set*) such that every pair $(x, y)$ with $x, y \in S$ and $x \neq y$ is a Li-Yorke pair.*[4]

The crucial ingredient of this analysis is the existence of periodic orbit of period 3.

**Theorem 3.9.** *If $b \in (0, 1) \setminus \{0.5\}$, then there exists $a_b$ such that if $a > a_b$ then $f_{a,b}$ has periodic orbit of period 3.*

By the Sharkovsky Theorem ([51], see also [37]), existence of a periodic orbit of period 3 implies existence of periodic orbits of all periods, and by the result of [37], it implies that the map is Li-Yorke chaotic. Thus, we get the following corollary:

**Corollary 3.10.** *If $b \in (0, 1) \setminus \{0.5\}$, then there exists $a_b$ such that if $a > a_b$ then $f_{a,b}$ has periodic orbits of all periods and is Li-Yorke chaotic.*

This result has a remarkable implication in non-atomic routing games. Recall that the parameter $a$ expresses the normalized total flow/demand; thus, Corollary 3.10 implies that when the game is asymmetric, i.e. when an interior equilibrium flow is not the $50\% - 50\%$ split, increasing the total demand of the system will inevitably lead to chaotic behavior, regardless of the form of the cost functions.[5]

# 4 Analysis of time-average regret

In the previous section, we discussed the time average convergence to Nash equilibrium for the map $f_{a,b}$. We now employ this property to investigate the time-average regret from learning with MWU.

**Theorem 4.1.** *The limit of the time-average regret is the total demand $N$ times the limit of the observable $(x - b)^2$ (provided this limit exists). That is*

$$\lim_{T \to \infty} \frac{R_T}{T} = N \left( \lim_{T \to \infty} \frac{1}{T} \sum_{n=1}^{T} (x_n - b)^2 \right). \tag{8}$$

Observe that if $x$ is a generic point of an ergodic invariant probability measure $\mu$, then the time limit of the observable $(x - b)^2$ is equal to its space average $\int_0^1 (x - b)^2 \, d\mu(x)$. This quantity is the variance of the random variable identity (we will denote this variable $X$, so $X(x) = x$) with respect to $\mu$. Typical cases of such a measure $\mu$, for which the set of generic points has positive Lebesgue measure, are when there exists an attracting periodic orbit $P$ and $\mu$ is the measure equidistributed on $P$, and when $\mu$ is an ergodic invariant probability measure absolutely continuous with respect to the Lebesgue measure [13]. In analogue to the family of quadratic interval maps [38], we have reasons to expect that for the Lebesgue almost every pair of parameters $(a, b)$ Lebesgue almost every point $x \in (0, 1)$ is generic for a measure of one of those two types.

**Upper bound for time-average regret**: Let $a > 4$ and $b \in (0, 1)$. Recall from (7) that $f_{a,b}$ has two critical points $x_l$ and $x_r$. Let $y_{\min} = f_{a,b}(x_r)$, $y_{\max} = f_{a,b}(x_l)$.

**Lemma 4.2.** *$I = [y_{\min}, y_{\max}]$ is invariant and globally absorbing on $(0, 1)$ for $a > \frac{1}{b(1-b)}$.*

Lemma 4.2 implies upper bounds on the variance of $x$ and thus on regret.

**Corollary 4.3.** *For $N > \frac{1}{b(1-b)}$ the time-average regret is bounded above by*

$$\lim_{T \to \infty} \frac{R_T}{T} = N \left( \lim_{T \to \infty} \frac{1}{T} \sum_{n=1}^{T} (x_n - b)^2 \right) \leq N(y_{\max} - b)(b - y_{\min}). \tag{9}$$

# 5 Analysis of time-average social cost

We begin this section with an extreme scenario of the time-average social cost; for a symmetric equilibrium flow ($b = 0.5$), the time-average social cost can be arbitrarily close to its worst possible

value. In contrast to the optimal social cost attained at the equilibrium $b = 0.5$, the long-time dynamics alternate between the two periodic points of the limit cycle of period two, which, at large population size, can approach 1 or 0 arbitrarily closely, see Figure 1 (top). This means almost all users will occupy the same route, while simultaneously alternating between the two routes.

**Theorem 5.1.** *For $b = 0.5$, the time-average social cost can be arbitrarily close to its worst possible value for a sufficiently large $a$, i.e. for a sufficiently large population size[6]. Formally, for any $\delta > 0$, there exists an $a$ such that, for any initial condition $x_0$, except countably many points whose trajectories eventually fall into the fixed point $b$, we have*

$$\liminf_{T \to \infty} \frac{1}{T} \sum_{n=1}^{T} SC(x_n) > \max_x SC(x) - \delta$$

More generally, when the equilibrium flow is asymmetric ($b \neq 0.5$), we can relate the normalized time-average social cost to the non-equilibrium fluctuations from the equilibrium flow. From (6):

$$\text{normalized time-average social cost} = \frac{\frac{1}{T} \sum_{n=1}^{T} \left( x_n^2 - 2\beta x_n + \beta \right)}{\beta(1 - \beta)} = 1 + \frac{\text{Var}(X)}{\beta(1 - \beta)}. \quad (10)$$

If the dynamics converges to the fixed point $b$, the variance vanishes and the normalized time-average social cost is optimal. However, as the total demand $N$ increases, the system suddenly bifurcates at $N = N_b^* \equiv 2/b(1 - b)$, which is the carrying capacity of the network. For $M > N_b^*$, the system is non-equilibrating, and the variance becomes positive. As a result, the normalized time-average social cost becomes suboptimal. For more details, see SM E.

## 6 Conclusion

Our benign-looking model of learning in routing games turns out to be full of surprises and puzzles. The dynamical system approach provides a useful framework to connect non-equilibrating dynamics, chaos theory, and topological entropy with standard game-theoretic (equilibrium) metrics such as regret and Price of Anarchy. Our results reveal a much more elaborate picture of learning in games than was previously understood. Exploring further upon these network of connections for different dynamics and games is a fascinating challenge for future work at the intersection of online optimization, game theory, dynamical systems, and chaos.

## Acknowledgements

Thiparat Chotibut and Georgios Piliouras acknowledge SUTD grant SRG ESD 2015 097, MOE AcRF Tier 2 Grant 2016-T2-1-170, grant PIE-SGP-AI-2018-01 and NRF 2018 Fellowship NRF-NRFF2018-07. Fryderyk Falniowski acknowledges the support of the National Science Centre, Poland, grant 2016/21/D/HS4/01798 and COST Action CA16228 "European Network for Game Theory". Research of Michał Misiurewicz was partially supported by grant number 426602 from the Simons Foundation.

## Broader societal impact

Our theoretical work provides a model that suggests that societal systems whose performance is impacted negatively under increased demand (e.g. road networks, public health services, etc.) might undergo violent phase transitions after exceeding critical thresholds. One could in principle use our quantitative predictions and toolsets to try to predict whether such a complex networked system is close to the onset of chaos/instability and to try to mitigate its destructive consequences.

## Footnotes

[1]For example in [30], in a class of payoff games called Median Effort, the payoff sensitivity parameter $a$ (referred to typically by $\lambda$ in the behavioral games literature) is estimated in different variants of EWA as being equal to 6.827 when only a single class of agents is considered. When allowing for two classes of agents, the best fit is found at $a_1 = 17.987$ and $a_2 = 2.969$. Numerous experimental results can be found in [29–31] as well as in the well known behavioral game theory textbook [12].

[2]It is easy to see that one can normalize the cost range $C$ to be $[0, 1]$, without loss of generality, by simultaneously updating appropriately $\epsilon' \in (0, 1)$, see Section 2. It is effectively the scaling down of costs without updating $\epsilon$ that leads to a constrained MWU model.

[3]For simplicity, let $N$ be an even number.

[4]Intuition behind this definition as well as other properties of chaotic behavior of dynamical systems are discussed in SM A.

[5]In fact this result can be strengthen, see SM A.

[6]Recall from (3) that $a = N \ln \left( \frac{1}{1-\epsilon} \right)$.

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
