[Supplementary Material]

# Supplementary Materials (SM)
The route to chaos in routing games:
When is price of anarchy too optimistic?

## A  Background Material on Dynamical Systems

This section familiarizes the reader with key concepts from dynamical systems necessary for this work, e.g., chaotic behavior, absolutely continuous invariant measures, topological entropy, and ergodic theorem.

It seems that there is no universally accepted definition of chaotic behavior of a dynamical system. Most definitions of chaos concern one of the following aspects:

- complex behavior of trajectories, such as Li-Yorke chaos;
- fast growth of the number of distinguishable orbits of length $n$, such as positive topological entropy;
- sensitive dependence on initial conditions, such as Devaney or Auslander-Yorke chaos;
- recurrence properties, such as transitivity or mixing.

In this article, the first two are crucial. Also, in the presence of chaos, studying precise single orbit dynamics can be intractable; we study the average behavior of trajectories instead. Thus, it is important to know whether the average converges. This is when ergodic theorems come into play.

### A.1  Li-Yorke chaos and topological entropy

The origin of the definition of Li-Yorke chaos (see Definition 3.8) is in the seminal Li and Yorke's article [37]. Intuitively orbits of two points from the scrambled set have to gather themselves arbitrarily close and spring aside infinitely many times but (if $X$ is compact) it cannot happen simultaneously for each pair of points. Why should a system with this property be chaotic? Obviously the existence of a large scrambled set implies that orbits of points behave in unpredictable, complex way. More arguments come from the theory of interval transformations, in view of which it was introduced. For such maps the existence of one Li-Yorke pair implies the existence of an uncountable scrambled set [34] and it is not very far from implying all other properties that have been called chaotic in this context, see e.g. [47]. In general, Li-Yorke chaos has been proved to be a necessary condition for many other "chaotic" properties to hold. A nice survey of properties of Li-Yorke chaotic systems can be found in [8].

A crucial feature of the chaotic behavior of a dynamical system is exponential growth of the number of distinguishable orbits. This happens if and only if the topological entropy of the system is positive. In fact positivity of topological entropy turned out to be an essential criterion of chaos [27]. This choice comes from the fact that the future of a deterministic (zero entropy) dynamical system can be predicted if its past is known (see [57, Chapter 7]) and positive entropy is related to randomness and chaos.

For every dynamical system over a compact phase space, we can define a number $h(f) \in [0, \infty]$ called the *topological entropy* of transformation $f$. This quantity was first introduced by Adler, Konheim and McAndrew [1] as the topological counterpart of metric (and Shannon) entropy.

For a given positive integer $n$ we define the $n$-th Bowen-Dinaburg metric on $X$, $\rho_n^f$ as

$$\rho_n^f(x,y) = \max_{0 \le i < n} dist(f^i(x), f^i(y)).$$

We say that the set $E$ is $(n, \varepsilon)$-separated if $\rho_n^f(x,y) > \varepsilon$ for any distinct $x, y \in E$ and we denote by $s(n, \varepsilon, f)$ the cardinality of the most numerous $(n, \varepsilon)$-separated set for $(X, f)$.

**Definition A.1.** *The topological entropy of $f$ is defined as*

$$h(f) = \lim_{\varepsilon \searrow 0} \limsup_{n \to \infty} \frac{1}{n} \log s(n, \varepsilon, f).$$

We begin with the intuitive explanation of the idea. Let us assume that we observe the dynamical system with the precision $\varepsilon > 0$, that is, we can distinguish any two points only if they are apart by at least $\varepsilon$. Then, after $n$ iterations we will see at most $s(n, \varepsilon, f)$ different orbits. If transformation $f$ is mixing points, then $s(n, \varepsilon, f)$ will grow. Taking upper limit over $n$ will give us the asymptotic exponential growth rate of number of (distinguishable) orbits, and going with $\varepsilon$ to zero will give us the quantity which can be treated as a measure of exponential speed, with which the number of orbits grow (with $n$).

Both positive topological entropy and Li-Yorke chaos are local properties; in fact, entropy depends only on a specific subset of the phase space and is concentrated on the set of so-called nonwandering points [10]. The question whether positive topological entropy implies Li-Yorke chaos remained open for some time, but eventually it was shown to be true; see [7]. On the other hand, there are Li-Yorke chaotic interval maps with zero topological entropy (as was shown independently by Smítal [52] and Xiong [58]). For deeper discussion of these matters we refer the reader to the excellent surveys by Blanchard [6], Glasner and Ye [28], Li and Ye [36] and Ruette's book [47].

**Entropy of $f_{a,b}$ is positive**: After this discussion we can show how entropy behaves for $f_{a,b}$. For any interval map, we have the following:

**Theorem A.2** ([39]). *For an interval map $f$, the following assertions are equivalent:*

   *i)  $f$ has a periodic point whose period is not a power of 2,*

   *ii) the topological entropy of $f$ is positive.*

Thus, Theorem A.2 combined with Corollary 3.10 strengthen the latter.

**Corollary A.3.** *If $b \in (0,1) \setminus \{1/2\}$, then there exists $a_b$ such that if $a > a_b$ then $f_{a,b}$ has periodic orbits of all periods, positive topological entropy and is Li-Yorke chaotic.*

**Calculating entropy**: In general, computing the entropy is not an easy task. However, in the context of interval maps, topological entropy can be computed quite straightforwardly — it is equal to the exponential growth rate of the minimal number of monotone subintervals for $f^n$.

**Theorem A.4** ([40]). *Let $f$ be a piecewise monotone interval map and, for all $n \geq 1$, let $c_n$ be the minimal cardinality of a monotone partition for $f^n$. Then*

$$h(f) = \lim_{n \to \infty} \frac{1}{n} \log c_n = \inf_{n \geq 1} \frac{1}{n} \log c_n.$$

Moreover, for piecewise monotone interval maps, the entropy computed with any partition into intervals, on which the map is monotone, is equal to the topological entropy [2, Prop. 4.2.3]. This gives us a way to understand what positive entropy of $f_{a,b}$ means from a game-theoretic perspective. For $a > \frac{1}{b(1-b)}$ the map $f_{a,b}$ is a bimodal map with two critical points $x_l, x_r$ (defined in (7)) and a (unique in $(0,1)$) equilibrium $b \in (x_l, x_r)$. Because $x$ is the probability of choosing the first strategy, we can say that if $x < x_l$ or $x > x_r$, then one of the strategies is overused and if $x$ is close to $b$, $x \in [x_l, x_r]$, then the system is approximately at equilibrium. Now, we can take a partition $\{[0, x_l), [x_l, x_r], (x_r, 1]\}$ into three intervals on which $f_{a,b}$ is monotone. For every $x \in [0, 1]$ and for every $n \geq 1$ we encode three events for the $n$-th iteration of $x$: $\mathbf{x}[n] = A$ if the system is approximately at equilibrium, that is if $f_{a,b}^n(x) \in [x_l, x_r]$; $\mathbf{x}[n] = B$ if the second strategy is overused, that is when $f_{a,b}^n(x) \in [0, x_l)$ and $\mathbf{x}[n] = C$ if the first strategy is overused, $f_{a,b}^n(x) \in (x_r, 1]$. This way for every $x \in [0, 1]$ we get an infinite sequence $\mathbf{x}$ on the alphabet $\{A, B, C\}$. Now, the fact that $h(f_{a,b}) > 0$ implies that the number of different blocks of length $n$, which we can observe looking at different $\mathbf{x}$ we generated this way, will grow exponentially.

## A.2  Invariant measures and ergodic theorem

We can also discuss a discrete dynamical system in terms of a measure preserving transformation defined on a probability space. This approach can handle not only purely mathematical concepts but also physical phenomena in nature. This subsection is devoted to invariant measures, absolutely continuous measures and the most fundamental idea in ergodic theory — the Birkhoff Ergodic Theorem, which states that with probability one the average of a function along an orbit of an ergodic transformation is equal to the integral of the given function.

**Definitions.** Let $(X, \mathcal{B}, \mu)$ be a probability space and $f : X \mapsto X$ be a measurable map. The measure $\mu$ is $f$-invariant (a map $f$ is $\mu$-invariant) if $\mu(f^{-1}E) = \mu(E)$ for every $E \in \mathcal{B}$. For an $f$-invariant measure $\mu$ we say that $\mu$ is ergodic ($f$ is ergodic) if for all $E \in \mathcal{B}$ if $f^{-1}E = E$ then $\mu(E) = 0$ or $1$. A measure $\mu$ is absolutely continuous with respect to Lebesgue measure if and only if for every set $E \in \mathcal{B}$ of zero Lebesgue measure $\mu(E) = 0$.

We can now state the ergodic theorem.

**Theorem A.5** (Birkhoff Ergodic Theorem). *Let $(X, \mathcal{B}, \mu)$ be a probability space. If $f$ is $\mu$-invariant and $g$ is integrable, then*

$$\lim_{n \to \infty} \frac{1}{n} \sum_{k=0}^{n-1} g(f^k(x)) = g^*(x)$$

*for some $g^* \in L^1(X, \mu)$ with $g^*(f(x)) = g^*(x)$ for almost every $x$. Furthermore if $f$ is ergodic, then $g^*$ is constant and*

$$\lim_{n \to \infty} \frac{1}{n} \sum_{k=0}^{n-1} g(f^k(x)) = \int_X g \, d\mu$$

*for almost every $x$.*

Lastly, why do absolutely continuous invariant measures matter? Computer-based investigations are widely used to gain insights into the dynamics of chaotic phenomena. However, one must exercise caution in the interpretation of computer simulations. Often, chaotic systems exhibit multiple ergodic invariant measures [26], but if the absolutely continuous measure with respect to Lebesgue measure exists, the averages of a given observable (function) along the orbits obtained from the computer simulations will be equal to the integral of this observable with respect to our measure [11]. Thus, the theoretical measure and the computational measure coincide in this work.

# B Main Figures

## B.1 Potential function, cobweb diagrams and time evolution

Figure 2: Population increase drives period-doubling instability and chaos. See discussion in SM B.1.

Although our congestion game has an associated convex potential function $\Phi_{a,b}(x) = \frac{N^2}{2}\left(\alpha x^2 + \beta(1-x)^2\right) = \frac{a^2}{2}\left((1-b)x^2 + b(1-x)^2\right)$ whose unique global minimum is the Nash

equilibrium $b$ (without loss of generality, we set $\alpha + \beta = 1$ and $\epsilon = 1 - 1/e$ so that $a = N$ and $b = \beta$), MWU at large $a$, or equivalently large $N$ with a fixed $\epsilon$, do not converge to the equilibrium, unlike a gradient-like update with a small step size. A line connecting $\Phi_{a,b}(x_n)$ to $\Phi_{a,b}(x_{n+1}) = \Phi_{a,b}(f_{a,b}(x_n))$ in the left column of Figure 2 is encoded with the color representing the timestep $n$. Later times are shown in red, while earlier times are shown in blue. Cobweb diagrams of the map $f_{a,b}$ are shown in the middle column, while the dynamics of the map are shown on the right column. From top to bottom, values of $a$ increase while $b$ remains fixed at $0.7$, demonstrating population size-driven instability. At these parameter values, the map $f_{a,b}$ is bimodal (blue curves in the middle column). For small $a$ (top row), the dynamics converge to the Nash equilibrium $b$. As $a$ increases, the dynamics converge to a period 2 attractor (second row), a period 4 attractor (third row), and a chaotic attractor (bottom row). As shown in Section 3, however, the time average of these orbits is exactly the Nash equilibrium $b$, represented by the horizontal green dashed lines on the right column. The initial condition here is set to $x_0 = 0.2$. The bifurcation diagram associated with $b = 0.7$ is shown in Fig. 3.

## B.2 Bifurcation diagram, regret and social cost (asymmetric case $b \neq 0.5$)

Figure 3: Bifurcation diagram, regret and social cost when the equilibrium $b \neq 0.5$. See discussion in SM B.2.

The top figure in Figure 3 demonstrates instability of the routing game driven by the increase in total demand $N$. The Nash equilibrium here is set to $b = 0.7$. As usual, we fix the learning rate $\epsilon = 1 - 1/e$ so that $a = N$ for simplicity. At small $N$, the dynamics converges toward the fixed point $b$, which is the Nash equilibrium. However, as $N$ exceeds the carrying capacity of $N_b^* = 2/b(1 - b)$, the Nash equilibrium becomes repelling and the dynamics no longer converge to it. The period-doubling route to chaos begins. Remarkably, the time-average of all orbits is exactly $b$, (the green line that tracks the center of masses of the blue orbits).

The middle figure shows the time-average regret $\frac{1}{T}R_T$ in purple. It *suddenly* becomes strictly positive at the first bifurcation, consistent with the prediction of (8) which states that the time-average regret is proportional to the fluctuations from the Nash equilibrium. The green line shows our upper bound on the time-average regret from Equation (9).

The bottom figure shows the normalized time-average social cost (i.e., time average social cost divided by optimum). It also *suddenly* becomes greater than 1 at the first bifurcation, consistent

with the prediction of Equation (6). Hence, the Price of Anarchy bound (of 1) is only a valid upper bound for the system inefficiency before the first bifurcation arises. The time average cost of the non-equilibrating dynamics is proportional to its fluctuations away from the Nash equilibrium. The rate at which the normalized time-average social cost increases above unity at the first bifurcation is depicted by the tangent line (dashed blue), which is calculated from Equation (16).

## B.3 Bifurcation diagram, regret and social cost (symmetric case $b = 0.5$)

Figure 4: Bifurcation diagram, regret and social cost when the equilibrium $b = 0.5$. See discussion in SM B.3.

The top figure in Figure 4 demonstrates the instability of the routing game driven by the increase in total demand $N$. Here, $\epsilon = 1 - 1/e$ so $a = N$ as usual. In this symmetric case, the capacity of the network, under which long-time dynamics equilibrate, is $N_b^* = 8$. Above the capacity, attracting periodic orbits of period 2 emerges.

In the middle figure the time-average regret is shown in the purple. It suddenly becomes strictly positive at the bifurcation. The regret bound also well approximates the actual values.

In the bottom figure the normalized time-average social cost also *suddenly* becomes greater than 1 at the bifurcation. Even in the symmetric case, the classic Price of Anarchy metric fails for $N > N_b^* = 8$.

## C Model discussion

One of the most well-known learning models in behavioral game theory is the Experienced Weighted Attraction (EWA) model [12, 29–31]. This is an extremely influential model in behavioral game theory with thousands of citations. The EWA model includes as a special case the MWU algorithm. EWA is effectively designed as a multi-parametric model that can incorporate a number of useful features of pre-existing learning models. The core of the EWA model is two variables which are updated after each round. The first variable $N(t)$ captures abstractly the number of "observation-equivalents" of past experience. This is not actually needed to capture MWU and it can be set equal to 1 by setting the $\rho$ parameter of EWA equal to zero. The second variable is $A^j(t)$ and captures the "attraction" of a strategy after period $t$. Although the details are not particularly important, this

is allowed to capture the accumulated payoff of all actions for the agent (e.g. by setting $\phi = 1$ and $\delta = 1$ in the EWA model whose full details are outside the scope of our paper). Finally, given these "attractions" the choice probabilities allow for logit probability response which exactly captures MWU where $1 + \epsilon = e^\lambda$. At least one empirical test seems to result into a model that is closely matching MWU run with a large $\lambda = 14.830$ (see last line of Table 5 in [30]). At this point it is important to note that our main reasoning for pointing out these connections between EWA and MWU is to show that standard normalizing assumptions for MWU that are indeed useful for proving regret bounds may be a bit too restrictive if we wish to capture a representative range of human behavior. Moreover, running MWU with large step sizes, although antithetical to the intuition coming from worst case regret bounds may indeed be a reasonable model at least in some cases of games and at least for some populations. It is perhaps not surprising that no single perfect value exists for any of the related parameters and indeed different games and possibly different groups of people are better captured by different parameters values (some large, some small). Although, we do not claim by any means that MWU with unconstrained values is a perfect or even always plausible model of human behavior, its analysis is an important step forward towards a theoretical treatment of much more detailed models such as the complete EWA model. Moreover, we believe that the fact that we can analyze MWU in this unconstrained setting using techniques that are orthogonal to the standard regret bounds can only help us understand MWU even more thoroughly than before.

Next we will address directly the issue of cost normalization and discuss why normalizing costs does not immediately address the issues in congestion games with many agents. Before doing so we should note that it is not necessary for MWU to be executed with costs lying in the $[0, 1]$ interval. MWU can be analyzed to show regret bounds that directly depend on the range of costs in the game which do not have to lie in $[0, 1]$ (analogously in the language of online convex optimization regret bounds that depend on the $L$-Lipschitz constants of the cost functions); see analysis of Normalized Exponentiated Gradient, which is equivalent to MWU, in [50] corollaries 2.14, 2.16 for more details.

To see why normalizing costs does not immediately address the issues of learning in congestion games with many agents, let's take costs $c_t$ normalized to be in $[0, 1]$ and then for any $\epsilon \in (0, 1)$ we have the standard regret bound:

$$\sum_{t=1}^{T} c_t(x_t) \le \frac{\ln(n)}{\epsilon} + \frac{1}{1 - \epsilon} \min_{x^* \in X} \sum_{t=1}^{T} c_t(x^*). \tag{11}$$

Let's focus on the term $\frac{\ln(n)}{\epsilon}$. This term is in units of maximum possible cost, e.g., what if all drivers nonsensically chose the same road. In standard online learning applications such as multi-armed bandits, Rock-Paper-Scissors, the intuition is that max unit cost per day is a penny and one can effectively discard this "day zero" cost term (i.e., $\frac{\ln(n)}{\epsilon}$) as noise. Let's call these low stakes games.

Congestion games with a large number of users are high stake games. As we keep increasing the number of users N the maximum possible cost will inevitably become prohibitively large. In such games bound (11) becomes impractical as there is no way to amortize this cost within a reasonable time horizon.

**Example:** Let $\epsilon = 0.01$ in a congestion game with $n = 10$ parallel edges/routes. Let the normalized cost functions on each edge $i$ be $\frac{x^4 + d}{N^4 + d}$ in accordance to the standard quartic cost model. The max cost of each agent is 1. The cost of each agent at the Nash equilibrium flow is approximately $(1/n)^4$ as $N$ grows large since the effect of the constant term $d$ dissipates. The ratio $\frac{\ln(n)}{\epsilon}$Max Cost/Nash cost = $\frac{\ln(n)}{\epsilon} n^4 = 2.3 \cdot 10^2 \cdot 10^4 = 2.3 \cdot 10^6$ days are needed to cover "day zero" costs in (11), i.e., the term $\frac{\ln(n)}{\epsilon}$ by paying an extra Nash equilibrium cost each day.

We are particularly interested in this regime. What would happen in the day-to-day behavior of a modern congested city in the next few years? This requires the usage of new tools such as dynamical systems analysis and experimental tests.

It is also important to note that the costs being in range $[0, M]$ instead of $[0, 1]$ is not really restrictive. One can always perform the following transformation that leaves MWU trajectories invariant. Scale cost down by a multiplicative cost $M$, $c' = c/M$, while using a new, larger $\epsilon' = 1 - (1 - \epsilon)^M$. This transformation could of course lead to a value of $\epsilon$ close to 1, which seems bad for bound (11). This

indeed would be an issue generally, but in our case we focus on high stakes settings where by design (11) is not practically applicable as our example above shows. Surprisingly, our results, in fact, show that this insistence on small learning rates is not even always necessary as we can exactly characterize settings where we converge to equilibria, with vanishing regret and optimal social cost despite using "large" $\epsilon$, where once again PoA techniques do not apply. Other recent papers have shown that this phenomenon arises not only in potential games but much more generally. For example [4] establishes finite regret with fixed step-sizes for alternating gradient descent dynamics in arbitrary two-agent games.

Normalizing costs also leads to other problems with no easy solutions as well. Let's for example consider the alternative normalizing cost model. E.g., when a driver is experiencing 20 minutes delay which lies in an interval $[0, M]$, then she should scale it down to $[0, 1]$ and then plug it into MWU with the original $\epsilon$, presumably something small and fixed/capped at e.g. $0.01$. This indeed is one possible model, but this is not the one we choose as in our setting this raises issues that cannot be easily addressed.

The first issue is how is this cost normalization implemented in practice? When a driver experiences 20 minutes cost in New York, what does she has to divide it with before entering it to MWU? Presumably her max cost to satisfy the $[0, 1]$ constraint. How can a user know this normalization factor?

A second issue has to do with the fact that the behavioral implications of the model are not always natural. The behavioral model suggested above would have users in NYC that experience 20 min delay divide this number with their maximum possible delay, which is some really large number in minutes as it corresponds to the black swan worst case scenario where all drivers use the same street. So effectively, the normalized costs that enter MWU are typically very small numbers e.g. $0.001$. With a fixed $\epsilon$ for instance at $0.01$ the agents are for all practical purposes unresponsive to traffic costs. Moreover, when an agent moves from a small city to a large city, they become more "relaxed" when it comes to their experienced delays as they scale down their actual costs more sharply.

Finally, a model where agents scale down costs by a multiplicative constant, when applied in a PoA setting, seems to be using simultaneously two conceptually inconsistent axioms. The first axiom is that the cost of games are not amenable to affine transformations, particularly payoff shifts, since these do not leave PoA invariant. As is well known, the PoA literature departs from the von Neumann and Morgenstern model of utilities and assumes that utilities exist as specific numbers in their given units. When it comes to learning, though, a second axiom is invoked and these utilities are automatically rescaled to $[0, 1]$. This is despite the fact that the behavior of all online learning dynamics e.g. GD, MWU, and even algorithms outside Follow-the-Regularized-Leader algorithms, such as Optimistic MWU, logit learning, etc, are *not* invariant under this payoff transformation. So, the affine payoff transformations that affect PoA analysis are not allowed, but the affine payoff transformations that affect learning are automatically enforced (by whom?). We believe that the effects of the scale of payoffs, learning rates should be carefully studied and our paper is an important step in this direction. Finally, our behavioral model fully encompasses the alternative as a special case, when the max game costs are indeed $1$.

We end by making one final note about how in the case of increasing population size not even shrinking step-sizes always suffice to avoid chaos. Our analysis for a fixed learning rate $\epsilon$ can easily be extended to capture non-equilibrating phenomena for arbitrary sequences of shrinking step sizes, as long as we allow for a dynamically evolving, increasing population. It should be already clear that the step-size $\epsilon$ and the population size $N$ (or equivalently the value of the maximum cost $M$) are competing forces that control system's stability. The larger population size implies the larger maximum cost $M$, which in turn implies the larger time horizon for MWU with a shrinking step-size algorithm to acquire smaller time-average regret, and for the classic equilibrium, Price of Anarchy, analysis to restore its predictive power. Unfortunately, if the population increases at a sufficiently fast rate to counter the shrinking step-size rate, the time-average regret will never vanish. Specifically, from our analysis, we proved that the relevant parameter that controls the long-time dynamics (e.g. equilibration, limit cycles, or chaos) and the social cost is $a = N \ln\left(\frac{1}{1-\epsilon}\right)$, see Equation 3. As long as at every time step $n$, $a(n) = N(n) \ln\left(\frac{1}{1-\epsilon(n)}\right)$ is greater than the chaotic threshold then the system will always remain in the chaotic regime despite the step-size going to zero. For example, for $\epsilon(n) = 1/\sqrt{n}$, it suffices that $N \geq \frac{a_b}{\ln\left(\frac{1}{1-1/\sqrt{n}}\right)}$ where $a_b$ is the threshold of chaos defined in

Theorem 3.9. Simple calculations show that it suffices $N \geq a_b \sqrt{n} \geq \frac{a_b}{\ln\left(\frac{1}{1-1/\sqrt{n}}\right)}$, Namely, a slowly (sublinearly) increasing population suffices for the system to remain forever in its non-equilibrating, inefficient, chaotic regime.

# D  Proofs

## D.1  Proof of Theorem 4.1

**Theorem 4.1.** *The limit of the time-average regret is the total demand $N$ times the limit of the observable $(x - b)^2$ (provided this limit exists). That is*

$$\lim_{T \to \infty} \frac{R_T}{T} = N \left( \lim_{T \to \infty} \frac{1}{T} \sum_{n=1}^{T} (x_n - b)^2 \right). \tag{12}$$

*Proof.* Recall that the time-average regret is

$$\frac{1}{T} R_T = \frac{1}{T} \sum_{n=1}^{T} \left( \alpha N x_n^2 + \beta N (1 - x_n)^2 \right) - \min \left\{ \frac{1}{T} \sum_{n=1}^{T} \alpha N x_n, \frac{1}{T} \sum_{n=1}^{T} \beta N (1 - x_n) \right\} \tag{13}$$

Consider

$$\frac{1}{T} \left( \sum_{n=1}^{T} \alpha x_n - \sum_{n=1}^{T} \beta (1 - x_n) \right) = \frac{1}{T} \sum_{n=1}^{T} [(\alpha + \beta) x_n - \beta] = (\alpha + \beta) \left( \frac{1}{T} \sum_{n=1}^{T} x_n - \frac{\beta}{\alpha + \beta} \right). \tag{14}$$

The quantity $\frac{\beta}{\alpha+\beta}$ is the system equilibrium $b$. Without loss of generality we assume as mentioned earlier $\alpha + \beta = 1$. Then, $b = \beta$ and $\lim_{T \to \infty} \left( \frac{1}{T} \sum_{n=1}^{T} x_n \right) = b$, by Theorem 3.3.

Therefore, in the limit $T \to \infty$, two terms in min-term of (13) coincide and we have by substituting $\alpha = 1 - \beta$ and remembering that $\beta = b$

$$\begin{aligned}
\lim_{T \to \infty} \frac{R_T}{T} &= \lim_{T \to \infty} \frac{N}{T} \sum_{n=1}^{T} \left( (1 - \beta) x_n^2 + \beta (1 - x_n)^2 - \beta (1 - \beta) \right) \\
&= \lim_{T \to \infty} \frac{N}{T} \sum_{n=1}^{T} \left( x_n^2 - 2\beta x_n + \beta^2 \right) = \lim_{T \to \infty} \frac{N}{T} \sum_{n=1}^{T} (x_n - b)^2.
\end{aligned}$$

$\square$

## D.2  Proof of Lemma 4.2

**Lemma 4.2.** *For $a > \frac{1}{b(1-b)}$ the interval $I = [y_{\min}, y_{\max}]$ is invariant and globally absorbing on $(0, 1)$.*

*Proof.* Fix $b \in (0, 1)$. Simple calculations show that $b \in (x_l, x_r)$ if and only if $a > 1/b(1 - b)$.[7] From now we assume that $a > 1/b(1 - b)$. Recall that $b$ is a fixed point of $f_{a,b}$ so we have $b = f_{a,b}(b) \in f_{a,b}([x_l, x_r]) = [y_{\min}, y_{\max}] = I$. Therefore $b \in I \cap (x_l, x_r)$ and $I \cap (x_l, x_r) \neq \emptyset$. Because $f_{a,b}$ is decreasing between the critical points, we have $f'_{a,b}(b) = ab^2 - ab + 1 < 0$. The latter and the uniqueness of a fixed point in $(0, 1)$ implies that $f_{a,b}(x) > x$ for $x \in (0, b)$, and $f_{a,b}(x) < x$ for $x \in (b, 1)$.

Obviously, if $x \in I \cap (x_l, x_r)$, then $f_{a,b}(x) \in f_{a,b}([x_l, x_r]) = I$. For $x \in [y_{\min}, x_l)$ we have $f_{a,b}(x) < f_{a,b}(x_l) = y_{\max}$. Suppose that $f_{a,b}(x) < y_{\min}$, then $f_{a,b}(x) < y_{\min} \leq x$ but it is impossible because $f_{a,b}(x) > x$ for $x \in (0, b)$. Thus $f_{a,b}([y_{\min}, x_l]) \subset I$. The same reasoning shows that $f_{a,b}([x_r, y_{\max}]) \subset I$. Thus $f_{a,b}(I) \subset I$.

Now let $x \in (0, x_l)$. Obviously $f_{a,b}(x) < y_{\max}$ and because $x_l < b$ we have $f_{a,b}(x) > x$ for $x < x_l$. To show that the orbit of $x$ falls eventually into $I$, it is sufficient to show that there exists $n$ such that $f_{a,b}^n(x) > y_{\min}$. Suppose that there is no such $n$, that is $f_{a,b}^n(x) < y_{\min}$ for all $n$. The sequence $(f_{a,b}^n(x))_{n \geq 1}$ is increasing and bounded, so it has a limit. Denote the limit by $c \in (0, x_l)$. Then $f_{a,b}(c)$ has to be equal $c$, but this contradicts the fact that on $(0, x_l)$ there is no fixed point of $f_{a,b}$. Thus, orbits of every point from $(0, x_l)$ will eventually fall into the invariant set $I$. Similar reasoning will show that orbits of every point from $x \in (x_r, 1)$ will eventually fall into $I$.

$\square$

### D.3 Proof of Theorem 5.1

**Theorem 5.1.** *For $b = 0.5$, the time-average social cost can be arbitrarily close to its worst possible value for a sufficiently large $a$, i.e. for a sufficiently large population size[8]. Formally, for any $\delta > 0$, there exists an $a$ such that, for any initial condition $x_0$, except countably many points, whose trajectories eventually fall into the fixed point $b$ we have*

$$\liminf_{T \to \infty} \frac{1}{T} \sum_{n=1}^{T} SC(x_n) > \max_x SC(x) - \delta$$

*Proof.* For a symmetric equilibrium $b = 0.5$, the two cost functions increase with the loads at the same rate $\alpha = \beta$. Recall from Section 2.2 that the social cost when fraction $x$ of the population adopts the first strategy is $SC(x) = \alpha N^2 x^2 + \alpha N^2 (1-x)^2$. This strictly convex function attains its minimum at the equilibrium $x = b = 0.5$, and its maximum of $\alpha N^2$ at $x = 0$ or $x = 1$. By Theorem 3.7 we know that for $a > 8$ there exists a periodic attracting orbit $\{\sigma_a, 1 - \sigma_a\}$, where $0 < \sigma_a < 0.5$. This orbit attracts trajectories of all points of $(0, 1)$, except countably many points, whose trajectories eventually fall into the repelling fixed point $0.5$. To establish that for all trajectories attracted by the orbit $\{\sigma_a, 1 - \sigma_a\}$ the time-average limit of the social cost can become arbitrarily close to $\alpha N^2$, it suffices to show that the distance of the two periodic points (of the unique attracting period-2 limit cycle) to the nearest boundary goes to zero as $a \to \infty$. Thus, it suffices to show that given any $\delta > 0$, there exist an $a$ such that $\sigma_a < \delta$.

For brevity, we denote the map $f_{a,0.5}$ by $f_a$. Then, since $\sigma_a$ is a periodic point of a limit cycle of period 2, we have $f_a^2(\sigma_a) = \sigma_a$. The last equality implies $f_a(\sigma_a) = 1 - \sigma_a$, which, after simple calculations, implies

$$\left(\frac{\sigma_a}{1 - \sigma_a}\right)^2 = \exp[a(\sigma_a - 0.5)].$$

Consider the function $\phi(x) = \left(\frac{x}{1-x}\right)^2 - \exp[a(x - 0.5)]$, then $\phi(0) = -\exp[-0.5a] < 0$. On the other hand, for any $\delta \in (0, 0.5)$, $\phi(\delta) = \left(\frac{\delta}{1-\delta}\right)^2 - \exp[a(\delta - 0.5)] > \frac{1}{2}\left(\frac{\delta}{1-\delta}\right)^2 > 0$ for a sufficiently large $a > 0$. The intermediate value theorem implies $\sigma_a \in (0, \delta)$, and the theorem follows. $\square$

## E   Analysis of variance spreading at the first period-doubling bifurcation

We study the behavior of the variance as $a$ crosses the period doubling bifurcation point. We first consider the model situation, where the map is $g(x) = (\gamma_1 - 1)x + \gamma_2 x^2 + \gamma_3 x^3$. Note that $\gamma_1$ and $\gamma_2$ here are not the coefficients of the cost functions. In this model situation, the bifurcation occurs at $\gamma_1 = 0$. If $\gamma_1 > 0$ then the fixed point $x = 0$ is attracting, and as $\gamma_1 < 0$ then it is repelling, but under some conditions on the coefficients there is an attracting periodic orbit of period 2. We are interested only at the limit behavior as $\gamma_1$ goes to zero, in a small neighborhood of $x = 0$. Therefore we may ignore all powers of $x$ larger than 3 and all powers of $\gamma_1$ larger than 1. Period 2 points are non-zero solutions of the equation

$$x = (\gamma_1 - 1)[(\gamma_1 - 1)x + \gamma_2 x^2 + \gamma_3 x^3] + \gamma_2[(\gamma_1 - 1)x + \gamma_2 x^2 + \gamma_3 x^3]^2 + \gamma_3[(\gamma_1 - 1)x + \gamma_2 x^2 + \gamma_3 x^3]^3.$$

Ignoring higher order terms and dividing by $x$, we get the equation

$$(2\gamma_2^2\gamma_1 - 2\gamma_2^2 + 4\gamma_1\gamma_3 - 2\gamma_3)x^2 - \gamma_1\gamma_2 x - 2\gamma_1 = 0.$$

Its discriminant is (after ignoring higher order terms in $\gamma_1$)

$$\Delta = -16\gamma_1(\gamma_2^2 + \gamma_3),$$

so

$$x = \frac{\gamma_1\gamma_2 \pm 4\sqrt{-\gamma_1(\gamma_2^2 + \gamma_3)}}{4(\gamma_2^2\gamma_1 - \gamma_2^2 + 2\gamma_1\gamma_3 - \gamma_3)}.$$

Assume now that $\gamma_2^2 + \gamma_3 > 0$. This is equivalent to $Sg(0) < 0$, so we will be able to apply it to our map (see Proposition 3.1). If $\gamma_1$ is close to zero, in the numerator $\gamma_1$ is negligible compared to $\sqrt{\gamma_1}$, and in the denominator $\gamma_1$ is negligible compared to a constant. Thus, approximately we have

$$x = \pm\sqrt{\frac{-\gamma_1}{\gamma_2^2 + \gamma_3}}.$$

Therefore, the variance is $\mathrm{Var}(X) = \frac{-\gamma_1}{\gamma_2^2 + \gamma_3}$.

After Taylor expanding the map of Equation (4) around the fixed point $b$ and comparing the coefficients to those of $g$, we obtain $\gamma_1 = 2 + ab(b-1)$, $\gamma_2 = a(b - \frac{1}{2})(1 + ab(b-1))$ and $\gamma_3 = a(1 + a(\frac{1}{6} + b(b-1))(3 + ab(b-1)))$. Recalling the first bifurcation occurs at $a_b^* = 2/b(1-b)$, we thus deduce the (right) derivative of the variance with respect to $a$ at the first period-doubling bifurcation:

$$\frac{d\mathrm{Var}(X)}{da}\bigg|_{a=a_b^{*+}} = -\frac{d\left(\frac{\gamma_1}{\gamma_2^2+\gamma_3}\right)}{da}\bigg|_{a=a_b^{*+}} = \frac{3b^3(1-b)^3}{2 - 6b(1-b)}, \tag{15}$$

which is a unimodal function in the interval $[0, 1]$ that is symmetric around $b = 0.5$, at which the maximum $0.09375$ is attained.

This allows us to deduce how fast the normalized time-average social cost increases at the first bifurcation, signalling how equilibrium Price of Anarchy metric fails as we increase $a$, or equivalently increase $N$. Namely, from Equation (6), one finds that the derivative of the normalized time-average social cost with respect to $a$ reads

$$\frac{d}{da}\,(\text{normalized time-average social cost})\,\bigg|_{a=a_b^{*+}} = \frac{1}{b(1-b)}\frac{d\mathrm{Var}(X)}{da}\bigg|_{a=a_b^{*+}} = \frac{3b^2(1-b)^2}{2 - 6b(1-b)}. \tag{16}$$

When $a < 2/b(1-b)$, the system equilibrates and the normalized time-average social cost is unity. However, when $a$ exceeds $2/b(1-b)$, the system is out of equilibrium, and normalized time-average social cost *suddenly* increases with a finite rate, given by Equation (16). At the first period-doubling bifurcation, the second-derivative with respect to $a$ becomes *discontinuous*, akin to the second order phase transition phenomena in statistical physics. Fig. 3 confirms the prediction of Equation (16).

Likewise, as the variance becomes positive, the time-average regret also becomes non-zero. At the first period-doubling bifurcation, the time-average regret given by Equation (8) *suddenly* increases with $a$ at the rate (where we use our typical normalization $a = N$)

$$\frac{d}{da}\,(\text{time-average regret})\,\bigg|_{a=a_b^{*+}} = \frac{d\,(a\mathrm{Var}(X))}{da}\bigg|_{a=a_b^{*+}} = \frac{3b^2(1-b)^2}{1 - 3b(1-b)}. \tag{17}$$

Therefore, at the first period-doubling bifurcation, where the equilibrium analysis begins to break-down, the following equality holds

$$\frac{d}{da}\,(\text{time-average regret})\,\bigg|_{a=a_b^{*+}} = 2\frac{d}{da}\,(\text{normalized time-average social cost})\,\bigg|_{a=a_b^{*+}}. \tag{18}$$

## F  Properties of attracting orbits

In this section, we investigate the properties of the attracting periodic orbits associated with the interval map $f_{a,b} : [0, 1] \to [0, 1]$

$$f_{a,b}(x) = \frac{x}{x + (1-x)\exp\left(a(x-b)\right)}. \tag{19}$$

We've argued in the main text that, when $b = 0.5$, the dynamics will converge toward the fixed point $b = 0.5$ whenever $a < 8$. And for any $a \geq 8$, the long-time dynamics will converge toward the attracting periodic orbits of period 2 located at $\{\sigma_a, 1 - \sigma_a\}$. The bifurcation diagram is thus symmetric around $b = 0.5$ as shown in the top picture of Fig. 4. In this case, the time-average regret is well-approximated by its upper bound, and the normalized time-average social cost asymptotes to the maximum value of 2.

When $b$ differs from $0.5$, we have argued in the main text that the emergence of chaos is inevitable, provided $a$ is sufficiently large. The period-doubling bifurcations route to chaos is guaranteed to arise. Fig. 1 of the main text shows chaotic bifurcation diagrams when $b = 0.7$. In this asymmetric case, standard equilibrium analysis only applies when the fixed point $b$ is stable, which is when $|f'_{a,b}(b)| \leq 1$, or equivalently when $a \leq 2/b(1-b)$.

Figure 5: Period diagrams of the small-period attracting periodic orbits associated with the map (19). The colors encode the periods of attracting periodic orbits as follows: period 1 (fixed point) = yellow, period 2 = red, period 3 = blue, period 4 = green, period 5 = brown, period 6 = cyan, period 7 = darkgray, period 8 = magenta, and period larger than 8 = white. The equilibrium analysis is only viable when the fixed point $b$ is stable, i.e. when $a \leq 2/b(1-b)$. In other region of the phase-space, non-equilibrating dynamics arise and system proceeds through the period-doubling bifurcation route to chaos in the white region. The picture is generated from the following algorithm: 20000 preliminary iterations are discarded. Then a point is considered periodic of period $n$ if $|f^n(x) - x| < 0.0000000001$ and it is not periodic of any period smaller than $n$. Slight asymmetry is caused by the fact that the starting point is the left critical point $x_l = 1/2 - \sqrt{1/4 - 1/a}$. In addition, for a fixed $a$, as we vary $b$ and penetrate into the chaotic regimes (white) from the outer layers, we numerically observe Feigenbaum's universal route to chaos as discussed below.

**Feigenbaum's universal route to chaos**: The period diagrams as a function of the two free-parameters $a$ and $b$ are shown in Fig. 5. It's interesting to report numerical observations of Feigenbaum's route to chaos for our bimodal map $f_{a,b}$. Although Feigenbaum's universality is known to apply among a one-dimensional *unimodal* interval map with a quadratic maximum [20, 35, 55], we also observe the Feigenbaum's period-doubling route to chaos for our *bimodal* interval map. Specifically, by fixing $a$ and varying $b$, we numerically measure the ratios

$$\delta_n \equiv \frac{b_{n+1} - b_n}{b_{n+2} - b_{n+1}}, \quad \alpha_n \equiv \frac{d_n}{d_{n+1}}, \tag{20}$$

where $b_n$ denotes the value at which a period $2^n$-orbits appears, and $d_n = f_{a,b}^{2^{n-1}}(x_l) - x_l$ such that the left critical point $x_l = \frac{1}{2}\left(1 - \sqrt{1 - \frac{4}{a}}\right)$ (the point at which $f_{a,b}$ attains its maximum) belongs to the $2^n$-orbits[9]. As $n$ grows large (we truncate our observation at $n = 12$), we find

$$\delta_{n=12} \approx 4.669\ldots, \quad \alpha_{n=12} \approx -2.502\ldots, \tag{21}$$

which agree, to 4 digits, with the Feigenbaum's universal constants, $\delta = 4.669201609102990\ldots$ and $\alpha = -2.502907875\ldots$, that appear, for example, in the period-doubling route to chaos in the logistic map.

**Coexistence of two attracting periodic orbits and non-uniqueness of regret and social cost**: The map $f_{a,b}$ has a negative Schwarzian derivative when $a > 4$, thus it has at most two attracting or neutral periodic orbits. Although the time-average of every periodic orbits converges *exactly* to the Nash equilibrium $b$, the variance $\lim_{T\to\infty} \frac{1}{T}\sum_{n=1}^{T}(x_n - b)^2$ of the coexisting periodic orbits can differ. Thus, the normalized time-average social cost and the time-average regret, which depend on the variance, can be multi-valued. Which value is attained depends on the variance of the attracting periodic orbits that the dynamics asymptotically reaches, which itself depends on the initial condition $x_0$. Period diagrams of Fig. 6 reveal how the two coexisting initial condition-dependent attracting periodic orbits are intertwined, and Fig. 7 reports the evidence of two coexisting periodic orbits whose variances differ, leading to multi-valued time-average regret and social cost.

Figure 6: Coexistence of two initial condition-dependent attracting periodic orbits. The pictures are generated from the same procedure as explained in Fig. 5, except that here the initial conditions for the top and the bottom pictures are located at the left and the right critical points, respectively. Also, $b \in [43/80, 51/80]$ and $a \in [4, 54]$. The color schemes are the same as those of Fig. 5 : period 1 (fixed point) = yellow, period 2 = red, period 3 = blue, period 4 = green, period 5 = brown, period 6 = cyan, period 7 = darkgray, period 8 = magenta, and period larger than 8 = white.

**Stability of the orbits**: In addition to the period diagrams, we investigate the stability of the attracting orbits by considering the Lyapunov exponents $\mathbb{E}\left(\log|f'_{a,b}|\right)$, where $\mathbb{E}(\cdot)$ denotes time-average. Fig. 8 (bottom) shows the Lyapunov exponents associated with different attracting orbits, revealing that extended-leg structures arise from the situations when the orbits become superstable, that is when one of the two critical points is an element of the orbits[10]. Within the regime of the same period (same color), there are situations when the two superstable extended-leg curves intersect. These scenarios happen when *both* critical points are elements of periodic orbits.

Figure 7: Coexistence of two attracting periodic orbits at $b = 0.61$ with two different variances implies *non-uniqueness* of time-average regret and normalized time-average social cost. As usual, we set $\epsilon = 1 - 1/e$ so that $N = a$. (Top) The range of $N$ in the shaded green region show coexistence of two attracting periodic orbits. The blue (red) periodic orbits is selected if the initial condition is the left (right) critical point $x_l$ ($x_r$). There are at most 2 coexisting attracting periodic orbits, as guaranteed by the negative Schwarzian derivative for our bimodal map $f_{a,b}$. The variance of the two periodic orbits are clearly different; thus, the time-average regret (middle) and the normalized time-average social cost (bottom) which depend on the variance are multi-valued. Which values are attained depend on initial conditions.

Also, note Fig. 5 reveals that the qualitatively similar extended-leg structures in the period diagrams appear in layers, with a chaotic regime sandwiched between two layers. Notice also that the consecutive layers have periods differ by 1. To understand why these layers with increasing periods appear, we investigate superstable periodic orbits in these layers and found that, all elements of the orbits, except for the left critical points $x_l = 1/2 - \sqrt{1/4 - 1/a}$ and its image $f_{a,b}(x_l)$, are approximately 0, independent of the period of the orbits. With this observation, we now approximate one of the superstable regions within each layer, using the time-average convergence to the Nash equilibrium property of Corollary 3.4. Namely, let $x_l$ be an element of a periodic orbit of period $p$ such that only $x_l$ and $f_{a,b}(x_l)$ are significantly larger than 0, then from Corollary 3.4 we have

$$x_l + f_{a,b}(x_l) + \underbrace{\left\{ f_{a,b}^2(x_l) + \cdots + f_{a,b}^{p-1}(x_l) \right\}}_{\approx 0} = pb. \tag{22}$$

Numerical results show that the approximation that every elements of the periodic orbits except $x_l$ and $f_{a,b}(x_l)$ are close to 0 becomes better and better for periodic orbits with larger periods; hence, we're interested in the limit of $a \gg 1$. To leading order in $\frac{1}{a}$, $x_l \approx \frac{1}{a}$ and $f_{a,b}(x_l) \approx \frac{1}{1+ae^{1-ab}}$ so that (22) gives $\frac{1}{a} + \frac{1}{1+ae^{(1-ab)}} \approx pb$.

Defining

$$S(a,b) = \frac{1}{ab} + \frac{1}{b + (ab)e^{(1-ab)}}, \tag{23}$$

we obtain the condition

$$S(a,b) \approx p, \tag{24}$$

Figure 8: (Bottom) Lyapunov exponents $\mathbb{E}\left(\log|f'_{a,b}|\right)$ numerically approximated by $\frac{1}{T}\sum_{n=1}^{T}\log|f'_{a,b}(x_n)|$ with $T = 2000$, shown in gray scale, superposed on the period diagrams (Top) adopted from Fig. 6 (Top). The color scheme of the Lyapunov exponents is such that $\mathbb{E}\left(\log|f'_{a,b}|\right) < -1.5$ is shown in white (very stable orbits) and $\mathbb{E}\left(\log|f'_{a,b}|\right) > 0$ is shown in black (unstable or chaotic). One can clearly see that the extended-leg structures arise from having superstable orbits as the skeleton of each attracting periodic orbits regime. When both critical points are elements of the attracting orbit, the two extended legs intersect. As expected, close to the bifurcation boundaries and in the chaotic regime, the orbits becomes unstable, as represented by the black color.

that should become more accurate as $a \gg 1$, for $x_l$ to be on the periodic orbit of period $p$ with the aforementioned property. Fig. 9 reveals that the level sets of $S(a,b)$ for $p = 2, 3, \ldots, 10$ accurately tracks the extended-leg structures with increasing periods, showing that these superstable orbits are the skeletons of the extended-leg layers shown in Fig. 5.

In addition, we can approximate the condition when *both* critical points $x_l$ and $x_r$ become the elements of these superstable periodic orbits. In these specific permutations of the orbits, we require $x_r = f_{a,b}(x_l)$. And from (22) we obtain $x_l + f_{a,b}(x_l) \approx pb$. Since $x_l + x_r = 1$, we conclude that both critical points will be on the periodic orbit of period $p$ with the aforementioned property when

$$b \approx \frac{1}{p}, \quad \text{and} \quad \frac{1}{a}\left[2\ln(a-1) + 1\right] \approx \frac{1}{p}, \tag{25}$$

where the condition on $a$ follows from (24) and $b \approx \frac{1}{p}$. Therefore, if we plot the relationship $b = \frac{1}{a}\left[2\ln(a-1) + 1\right]$, the graph will encompass the situations when both critical points are on the periodic orbits with the aforementioned property. This is illustrated by the dashed olive green line of Fig. 9 that passes through the intersections between two superstable curves within each period-$p$ region.

Figure 9: Layers of extended-leg structures with increasing periods arise from specific permutations of superstable periodic orbits. See discussion in text.

**Discussion about figure 9:** As argued in the stability of the orbits section, $S(a, b) = p$ defines the superstable periodic orbits of period $p$ with the property that only $x_l$ and $f(x_l)$ are the only two elements of the periodic orbits that are not near 0. The level sets of $S(a, b)$ at $p = 2, 3, \ldots, 10$ are displayed in different colors (bottom), which accurately track extended-leg curves with large negative Lyapunov exponents (top). The color scheme of the Lyapunov exponents is such that $\mathbb{E}\left(\log |f'_{a,b}|\right) < -1.5$ is shown in white (very stable orbits) and $\mathbb{E}\left(\log |f'_{a,b}|\right) > 0$ is shown in black (unstable or chaotic). The dashed olive green curve $b = \frac{1}{a}\left[2\ln(a-1) + 1\right]$ obtained from (25) encompasses the situations when *both* critical points are the only two non-near-zero elements of the periodic orbits, i.e. when the two superstable curves in each region of the same period $p$ intersect. These results provide a reasonable answer to why layers of extended-leg structures with increasing periods appear in the period diagram of Fig. 5.

## G  Extensions to congestion games with many strategies

In this section we extend our results on Li-Yorke chaos and time-average convergence to Nash equilibrium to the case of many strategies.

We will consider a $m$-strategy *congestion game* with a continuum of players/agents, where all of them use *multiplicative weights update*. Each of the players controls an infinitesimally small fraction of the flow. We will assume that the total flow of all the agents is equal to $N$. We will denote the fraction of the players using $i$ one of the $m$ strategies at time $n$ as $x_i(n)$ where $i \in \{1, \ldots, m\}$. Intuitively, this

model captures how a large population of players chooses between multiple alternative, parallel paths for going from point $A$ to point $B$. If a large fraction of the players choose the same strategy, this leads to congestion/traffic, and the cost increases. We will assume that the cost is proportional to the *load*. If we denote by $c(i)$ the cost of any player playing strategy number $i$, and the coefficients of proportionality are $\alpha_i$, then we get

$$c(i) = \alpha_i N x_i \ \forall i \in \{1, \ldots, m\}. \tag{26}$$

At time $n+1$ the players know already the cost of the strategies at time $n$ and update their choices. Since we have a continuum of agents we will assume that the fractions of users using the first, second, $m$-th path are respectively equal to the probabilities $x_1(n), \ldots, x_m(n)$. Once again we will update the probabilities using MWU. The update rule in the case of $m$ strategies is as follows:

$$x_i(n+1) = x_i(n) \frac{(1-\epsilon)^{c(i)}}{\sum_{j \in \{1,\ldots,m\}} x_j(n)(1-\epsilon)^{c(j)}}, \tag{27}$$

The (Nash) equilibrium flow $(b_1, \ldots, b_m)$ of the congestion games is the unique flow such that the cost of all paths are equal to each other. Specifically, the equilibrium is defined by $b_i = \frac{1/\alpha_i}{\sum_{j\{1,\ldots,m\}} \alpha_j}$.

The MWU dynamics introduced by (27) can be interpreted as the dynamics of the map $f$ of the simplex $\Delta = \{(x_1, \ldots, x_m) : x_i \geq 0, \sum_{j=1}^m x_j = 1\}$ to itself, given by

$$f(x_1, \ldots, x_m) = \left( \frac{y_1}{Y}, \ldots, \frac{y_m}{Y} \right), \tag{28}$$

where

$$y_i = x_i \exp(-a_i x_i) \text{ with } a_i = N\alpha_i \ln\left(\frac{1}{1-\epsilon}\right) \text{ and } Y = \sum_{j=1}^m y_j.$$

We set $a_i = Np_i$, $p_i = \alpha_i \ln(1/(1-\varepsilon))$, and see what happens as $N \to \infty$. We will show that if $N$ is sufficiently large then $f$ is Li-Yorke chaotic, except when $m=2$ and $p_1 = p_2$, that is $\alpha_1 = \alpha_2$.

### G.1 Proof of the existence of Li-Yorke chaos

In this section we will provide a generalization of Theorem 3.9 showing that even in congestion games with many strategies, increasing the population/flow will result to instability and chaos. Thus, the emergence of chaos is robust both in the size of the game (holds in games with few as well as many paths/strategies) as well as to the actual cost functions on the edges (chaos emerges for effectively any tuple of linear cost functions).

**Theorem G.1.** *Given any non-atomic congestion game with $m$ actions as described by model (26), (27), except for the case[11] of $m=2$ with $\alpha_1 = \alpha_2$, then there exists a total system demand $N_0$ such that for if $N \geq N_0$ the system has periodic orbits of all periods, positive topological entropy and is Li-Yorke chaotic.*

*Proof.* Set

$$p = \frac{1}{\sum_{k=1}^{m-1} \frac{1}{p_k}}$$

and consider the segment

$$I = \left\{ (x_1, \ldots, x_m) \in \Delta : x_i = \frac{px}{p_i} \text{ for } i < m, \ 0 \leq x \leq 1 \right\}.$$

We have

$$x_m = 1 - \sum_{k=1}^{m-1} x_k = 1 - x,$$

so indeed, $I \subset \Delta$.

We have $y_i = \frac{px}{p_i} \exp(-Npx)$ for $i < m$ and

$$y_m = (1-x)\exp(-Np_m(1-x)),$$

so

$$Y = x\exp(-Npx) + (1-x)\exp(-Np_m(1-x)).$$

Therefore, for $i < m$ we get

$$\frac{y_i}{Y} = \frac{\frac{px}{p_i}\exp(-Npx)}{x\exp(-Npx) + (1-x)\exp(-Np_m(1-x))}$$
$$= \frac{p}{p_i} \cdot \frac{x}{x + (1-x)\exp(-Np_m(1-x) + Npx)}.$$

Thus, $f(I) \subset I$, and the map $f$ on $I$ (in the variable $x$) is given by the formula

$$x \mapsto \frac{x}{x + (1-x)\exp(-Np_m(1-x) + Npx)}.$$

This formula can be rewritten as

$$x \mapsto \frac{x}{x + (1-x)\exp\left(N(p+p_m)\left(x - \frac{p_m}{p+p_m}\right)\right)},$$

and we already know that this map is Li-Yorke chaotic and has positive topological entropy and periodic orbits of all possible periods for sufficiently large $N$, provided $\frac{p_m}{p+p_m} \neq \frac{1}{2}$. Therefore, in this case $f$ is Li-Yorke chaotic and has positive topological entropy on the whole $\Delta$ for sufficiently large $N$.

Let us now investigate the exceptional case $\frac{p_m}{p+p_m} = \frac{1}{2}$. Then $p_m = p$, so

$$\frac{1}{p_m} = \sum_{k=1}^{m-1} \frac{1}{p_k}.$$

Therefore,

$$\frac{2}{p_m} = \sum_{k=1}^{m} \frac{1}{p_k}.$$

However, our choice of $m$ as a special index was arbitrary, so the only case when we do not get Li-Yorke chaos and positive topological entropy is when

$$\frac{2}{p_i} = \sum_{k=1}^{m} \frac{1}{p_k}$$

for every $i = 1, 2, \ldots, m$. In this case, $p_1 = p_2 = \cdots = p_m$, so we get $\frac{2}{p_i} = \frac{m}{p_i}$, so $m = 2$ and $p_1 = p_2$. Eventually $p_1 = p_2$ if and only if $\alpha_1 = \alpha_2$. $\qquad \square$

### G.2   Time average convergence to equilibrium

The goal in this section is to generalize Theorem 3.3 about the time-average of the flow converging to the (Nash) equilibrium flow. We will start with a technical lemma showing that the all (interior) initial conditions converge to an interior invariant set. We consider the map $f$ given by (28). If $x = (x_1, \ldots, x_m) \in \Delta$, then we write $\xi(x) = \min(x_1, \ldots, x_m)$.

**Lemma G.2.** *If $x \in \Delta$ and $\xi(x) > 0$ then*

$$\inf_{n \geq 0} \xi(f^n(x)) > 0. \tag{29}$$

*Proof.* We use notation from the definition of $f$. Set

$$A = \max(a_1, \ldots, a_m), \quad \alpha = \min(a_1, \ldots, a_m).$$

Since for every $i$ we have $0 \leq x_i \leq 1$, we get

$$y_i \geq x_i \exp(-Ax_i) \geq x_i \exp(-A). \tag{30}$$

There exists $k$ such that $x_k \geq 1/m$. Then $y_k \leq x_k \exp(-\alpha/m)$, so

$$x_k - y_k \geq \frac{1}{m}\left(1 - \exp\left(-\frac{\alpha}{m}\right)\right).$$

Since $x_j \geq y_j$ for all $j$, we get

$$1 - Y \geq \frac{1}{m}\left(1 - \exp\left(-\frac{\alpha}{m}\right)\right).$$

Set

$$C = \frac{1}{m}\left(m - 1 + \exp\left(-\frac{\alpha}{m}\right)\right).$$

Then $Y \leq C$ and $C < 1$. By (30), for every $i$ we get

$$\frac{y_i}{Y} \geq \frac{x_i \exp(-Ax_i)}{C} \geq \frac{x_i \exp(-A)}{C}.$$

We have

$$\lim_{t \to 0} \frac{\exp(-At)}{C} = \frac{1}{C} > 1,$$

so there is $\varepsilon > 0$ such that $x_i \leq \varepsilon$ then $y_i/Y \geq x_i$. Moreover, if $x_i \geq \varepsilon$ then $y_i/Y \geq \varepsilon \exp(-A)/C$. This proves that (29) holds. $\qquad\square$

We are ready to prove that the time average of the flow converges to the equilibrium flow. The update rule in the case of $m$ strategies is given by (27).

**Theorem G.3.** *Given any non-atomic congestion game with $m$ actions as described by model (26),(27), if $x = (x_1, \ldots, x_m) \in \Delta$, and $\min(x_1, \ldots, x_m) > 0$, then*

$$\lim_{T \to \infty} \frac{1}{T} \sum_{n=0}^{T-1} x_i(n) = b_i. \tag{31}$$

*where $(b_1, \ldots, b_m)$ is the (Nash) equilibrium flow of the congestion game, i.e., $b_i = \frac{1/\alpha_i}{\sum_{j\{1,\ldots,m\}} \alpha_j}$.*

*Proof.* By dividing two equations of type (27) one for strategy $i$ and one for strategy $j$ we derive that:

$$\frac{x_{n+1}(i)}{x_{n+1}(j)} = \frac{x_n(i)}{x_n(j)}(1-\epsilon)^{c_n(i)-c_n(j)} \tag{32}$$

By unrolling this relationship we derive:

$$\frac{x_{n+1}(i)}{x_{n+1}(j)} = \frac{x_1(i)}{x_1(j)}(1-\epsilon)^{\sum_{\tau=1}^{n}\left(c_\tau(i)-c_\tau(j)\right)} \tag{33}$$

By Lemma G.2, given any initial condition $(x_1(1), x_1(2), \ldots, x_1(n))$ such that $\min_i x_1(i) > 0$ we have that there exists $\delta > 0$ such that $\inf_{n \geq 0} \min x_n(i) > \delta$ and $\sup_{n \geq 0} \max x_n(i) < 1 - \delta$. Then

$$\frac{\delta}{1-\delta}\frac{x_1(j)}{x_1(i)} < (1-\epsilon)^{\sum_{\tau=1}^{n}\left(c_\tau(i)-c_\tau(j)\right)} < \frac{1-\delta}{\delta}\frac{x_1(j)}{x_1(i)}$$

and

$$\frac{1}{\ln(1-\epsilon)}\ln\left(\frac{\delta}{1-\delta}\frac{x_1(j)}{x_1(i)}\right) < \sum_{\tau=1}^{n}\left(c_\tau(i) - c_\tau(j)\right) < \frac{1}{\ln(1-\epsilon)}\ln\left(\frac{1-\delta}{\delta}\frac{x_1(j)}{x_1(i)}\right).$$

Dividing all sides of the inequality by $n$ we get

$$\frac{\frac{1}{\ln(1-\epsilon)}\ln\left(\frac{\delta}{1-\delta}\frac{x_1(j)}{x_1(i)}\right)}{n} < \frac{\sum_{\tau=1}^{n}\left(c_\tau(i) - c_\tau(j)\right)}{n} < \frac{\frac{1}{\ln(1-\epsilon)}\ln\left(\frac{1-\delta}{\delta}\frac{x_1(j)}{x_1(i)}\right)}{n}.$$

By taking limits we have that for any $i, j$ $\lim_n \frac{\sum_{\tau=1}^n \left( c_\tau(i) - c_\tau(j) \right)}{n} = 0$. For any subsequence such that the limits $\lim_n \frac{\sum_{\tau=1}^n c_\tau(i)}{n}$ exist for all $i$ we have that:

$$\lim_{n \to \infty} \frac{\sum_{\tau=1}^n c_\tau(i)}{n} = \lim_{n \to \infty} \frac{\sum_{\tau=1}^n \left( c_\tau(i) - c_\tau(j) \right) + \sum_{\tau=1}^n c_\tau(j)}{n} = \lim_{n \to \infty} \frac{\sum_{\tau=1}^n c_\tau(j)}{n} \qquad (34)$$

Since the cost functions are linear, i.e., $c_n(j) = a_j N x_n$, equation (34) implies that:

$$a_i \lim_{n \to \infty} \frac{\sum_{\tau=1}^n x_\tau(i)}{n} = a_j \lim_{n \to \infty} \frac{\sum_{\tau=1}^n x_\tau(j)}{n} \qquad (35)$$

and the point $\left( \lim_n \frac{\sum_{\tau=1}^n x_\tau(1)}{n}, \ldots, \lim_n \frac{\sum_{\tau=1}^n x_\tau(m)}{n} \right)$ is the unique equilibrium flow of the congestion game. Clearly, the same argument can be made for any other subsequence such that $\lim_n \frac{\sum_{\tau=1}^n c_\tau(i)}{n}$ exists by possibly defining its own subsequence so that all the other limits also exist (which is always possible due to compactness). By (34), (35) the value must once again agree with the unique equilibrium of the game. Hence, for any $i$, $\lim_n \frac{\sum_{\tau=1}^n c_\tau(i)}{n}$ exists and $\left( \lim_n \frac{\sum_{\tau=1}^n x_\tau(1)}{n}, \ldots, \lim_n \frac{\sum_{\tau=1}^n x_\tau(m)}{n} \right)$ is the unique equilibrium flow. $\qquad \square$

# H    Extensions to congestion games with polynomial costs

For simplicity, we will return to the case with exactly two actions/paths. As usual we will denote by $c(j)$ the cost of selecting the strategy number $j$ (when $x$ fraction of the agents choose the first strategy). We will focus on cost functions which are monomials with the same degree[12]

$$c_1(x) = \alpha N^p x^p \qquad\qquad c_2(x) = \beta N^p x^p, \qquad\qquad p \in \mathbb{N}. \qquad (36)$$

The (Nash) equilibrium flow corresponds to the unique split $(x_b^*, 1 - x_b^*)$ such that $c_1(x_b^*) = c_2(x_b^*)$.

## H.1    Multiplicative weights with polynomial costs

Once again, applying the multiplicative weights updates rule we get formula (2). By substituting into (2) the values of the polynomial cost functions from (36) we get:

$$\begin{aligned} x_{n+1} &= \frac{x_n (1-\epsilon)^{\alpha N^p x_n^p}}{x_n (1-\epsilon)^{\alpha N^p x_n^p} + (1-x_n)(1-\epsilon)^{\beta N^p (1-x_n)^p}} \\ &= \frac{x_n}{x_n + (1-x_n)(1-\epsilon)^{N^p (\beta(1-x_n)^p - \alpha x_n^p)}}. \end{aligned} \qquad (37)$$

We introduce the new variables

$$a = (\alpha + \beta) N^p \ln \left( \frac{1}{1-\epsilon} \right), \quad b = \frac{\beta}{\alpha + \beta}. \qquad (38)$$

Once again, we see that $b = 1/2$ if and only if the two paths are totally symmetric (same cost function). In this case $x_b^* = 1/2$ as well as the equilibrium flow splits the total demand equally in both paths.

We will thus study the dynamical systems generated by the one-dimensional map:

$$f_{a,b}(x) = \frac{x}{x + (1-x) \exp(a P_b(x))}. \qquad (39)$$

Clearly $f_{a,b} : [0,1] \to [0,1]$, where $0 < b < 1$, $a > 0$, and $P_b(x) = (1-b)x^n - b(1-x)^n$. We have $P_b(0) = -b$, $P_b(1) = 1 - b$, and $P_b$ is strictly increasing. Therefore, there exists the unique point $x_b^* \in (0,1)$ such that $P_b(x_b^*) = 0$. Observe that $f_{a,b}(x_b^*) = x_b^*$ is exactly the equilibrium flow. Moreover

$$f'_{a,b}(x) = \frac{(1 - ax(1-x)P'_b(x))\exp(aP_b(x))}{(x + (1-x)\exp(aP_b(x)))^2}. \tag{40}$$

From (40) we have that

$$f'_{a,b}(0) = \exp(-aP_b(0)) = \exp(ab) > 1 \quad \text{and} \quad f'_{a,b}(1) = \exp(aP_b(1)) = \exp(a(1-b)) > 1.$$

Thus 0 and 1 are repelling. This fact implies (the proof of this fact is the same as of Lemma 3.1 from [14]) that there exists an invariant attracting subset $I_{a,b}$ of the unit interval.

Although the time-average convergence of the flow does not necessarily converge to the equilibrium flow, one can still prove a theorem analogous to Theorem 3.3 that reflects the *time-average of the costs* of different paths. Informally, although the traffic flows through each path evolve chaotically from day-to-day, if an outsider observer was to keep track of their time-average cost, all paths would appear to be experience similar delays. It is due to this inability to learn a preferred path that chaos (as we will argue next) is self-sustaining even under polynomial cost functions despite the application of learning/optimizing dynamics.

**Theorem H.1.** *If cost functions $c_1$, $c_2$ are given by* (36), *then*

$$\lim_{n \to \infty} \frac{1}{n} \sum_{k=0}^{n-1} (c_1(x_k) - c_2(x_k)) = 0. \tag{41}$$

*Proof.* There is a closed interval $I_{a,b} \subset (0,1)$ which is invariant and attracting for $f_{a,b}$. Thus, there is $\delta \in (0,1)$ such that $I_{a,b} \subset (\delta, 1 - \delta)$.

Fix $x = x_0 \in [0,1]$ and use our notation $x_n = f_{a,b}^n(x_0)$. By induction we get

$$x_n = \frac{x}{x + (1-x)\exp\left[a\sum_{k=0}^{n-1}(c_1(x_k) - c_2(x_k))\right]}. \tag{42}$$

Assume that $x = x_0 \in I_{a,b}$. Since $\delta < x_n < 1 - \delta$, we have

$$\frac{x}{1-\delta} < x + (1-x)\exp\left(a\sum_{k=0}^{n-1}(c_1(x_k) - c_2(x_k))\right) < \frac{x}{\delta},$$

so

$$\delta^2 < x\frac{\delta}{1-\delta} < (1-x)\exp\left(a\sum_{k=0}^{n-1}(c_1(x_k) - c_2(x_k))\right) < x\frac{1-\delta}{\delta} < \frac{1}{\delta}.$$

Therefore

$$\delta^2 < \exp\left(a\sum_{k=0}^{n-1}(c_1(x_k) - c_2(x_k))\right) < \frac{1}{\delta^2}, \tag{43}$$

so

$$\left|a\sum_{k=0}^{n-1}(c_1(x_k) - c_2(x_k))\right| < 2\log(1/\delta).$$

This inequality can be rewritten as

$$\left|\frac{1}{n}\sum_{k=0}^{n-1}(c_1(x_k) - c_2(x_k))\right| < \frac{2\log(1/\delta)}{an},$$

and (41) follows.

If $x \in (0,1) \setminus I_{a,b}$, then by the definition of $I_{a,b}$ there is $n_0$ such that $f_{a,b}^{n_0}(x) \in I_{a,b}$, so (41) also holds. $\qquad\square$

## H.2 Proof of the existence of Li-Yorke chaos

**Theorem H.2.** *For any $b \in (0, 1/2) \cup (1/2, 1)$ there exists $a_0$ such that if $a > a_0$ then $f_{a,b}$ given by (39) has a periodic orbit of period 3, and therefore it has periodic orbits of all periods, positive topological entropy and is Li-Yorke chaotic.*

*Proof.* Fix $b \in (0, 1/2)$. It is enough to show that if $a$ is sufficiently large, then there exist $x_0, x_1, x_2, x_3$ such that $f_{a,b}(x_i) = x_{i+1}$ and $x_3 < x_0 < x_1$.

Our points $x_i$ will depend on $a$. We start by taking

$$x_1 = 1 - \frac{1}{a} \text{ and } y = \frac{x_b^*}{2}.$$

Note that $y$ does not depend on $a$ and $P_b(y) < 0$. The inequality $f_{a,b}(y) > x_1$ is equivalent to

$$y > (a-1)(1-y)\exp(aP_b(y)),$$

which holds for sufficiently large $a$. Moreover, for sufficiently large $a$ we have $f_{a,b}(x_b^*) < x_1$. Therefore, for sufficiently large $a$ there exists $x_0 \in (y, x_b^*)$ such $f_{a,b}(x_0) = x_1$. In particular, we have $x_0 < x_1$.

Set

$$b^* = \frac{3}{4} - \frac{b}{2}.$$

Since $b < 1/2$, we have $b^* < 1 - b$, so if $a$ is sufficiently large, then $P_b(x_1) > b^*$, and thus

$$x_2 = f_{a,b}(x_1) = \frac{x_1}{x_1 + \frac{1}{a}\exp(aP_b(x_1))} \leq \frac{a}{\exp(ab^*)} = a\exp(-ab^*).$$

Since $P_b(x_2) \geq -b$, we get

$$x_3 = f_{a,b}(x_2) = \frac{x_2}{x_2 + (1-x_2)\exp(aP_b(x_2))} \leq \frac{x_2}{x_2 + (1-x_2)\exp(-ab)}$$

$$= \frac{x_2\exp(ab)}{x_2\exp(ab) + 1 - x_2} \leq x_2\exp(ab) \leq a\exp(a(b-b^*)).$$

Since

$$b - b^* = b - \frac{3}{4} + \frac{b}{2} = \frac{3(b - \frac{1}{2})}{2} < 0,$$

we have $\lim_{a \to \infty} a\exp(a(b-b^*)) = 0$, and therefore if $a$ is sufficiently large, then $x_3 < y < x_0$. Hence, $f_{a,b}$ has a periodic orbit of period 3.

We have $f_{a,1-b}(1-x) = 1 - f_{a,b}(x)$, so $f_{a,1-b}$ is conjugate to $f_{a,b}$. Therefore, the theorem holds also for $b \in (1/2, 1)$. $\square$

We did not use too many properties of $P_b$, so the theorem holds for a larger class of those functions.

**Corollary H.3.** *Given any non-atomic congestion game with polynomial cost functions described by model (37), except for the symmetric case with $\alpha = \beta$, then there exists a total system demand $N_0$ such that for if $N \geq N_0$ the system has periodic orbits of all periods, positive topological entropy and is Li-Yorke chaotic.*

# I  Extensions to congestion games with heterogeneous users

This is the model for the case of heterogeneous population. We will start with the simplest possible case where there are only two subpopulations. We will consider a two-strategy *congestion game* with two continuums of players/agents, where all of them use *multiplicative weights update*. Each of the players controls an infinitesimal small fraction of the flow. Out of the total flow/demand $N$ of the first population has size $N\eta_1$ whereas the total flow of the second population is $N\eta_2$. A canonical example would be $\eta_1 = \eta_2 = 0.5$.

We will denote the fraction of the players of the first (resp. second) population using the first strategy at time $n$ as $x_n$ (resp. $y_n$). The second strategy is chosen by $1 - x_n$ (resp. $1 - y_n$) fraction of the

players. Intuitively, this model captures how two large population of players/cars (e.g. taxis versus normal cars) chooses between two alternative, parallel paths for going from point $A$ to point $B$. If a large fraction of the players choose the same strategy, this leads to congestion/traffic, and the cost increases. We will assume that the cost is proportional to the *load*. If we denote by $c(j)$ the cost of the player playing the strategy number $j$, and the coefficients of proportionality are $\alpha, \beta$, then we get

$$c(1) = \alpha N(\eta_1 x + \eta_2 y), \quad c(2) = \beta N(\eta_1 + \eta_2 - \eta_1 x - \eta_2 y) \tag{44}$$

For *multiplicative weights update* (MWU), for the first (resp. second), there is a parameter $\epsilon_1 \in (0,1)$, (resp. $\epsilon_2 \in (0,1)$) which can be treated as the common learning rate of all players of that population. Thus, we get

$$
\begin{aligned}
x_{n+1} &= \frac{x_n(1-\epsilon_1)^{c(1)}}{x_n(1-\epsilon_1)^{c(1)} + (1-x_n)(1-\epsilon_1)^{c(2)}}, \\
y_{n+1} &= \frac{y_n(1-\epsilon_2)^{c(1)}}{y_n(1-\epsilon_2)^{c(1)} + (1-y_n)(1-\epsilon_2)^{c(2)}}.
\end{aligned}
\tag{45}
$$

By combining equations (44) and (45),

$$
\begin{aligned}
x_{n+1} &= \frac{x_n}{x_n + (1-x_n)(1-\epsilon_1)^{N(\beta(\eta_1+\eta_2) - (\alpha+\beta)(\eta_1 x_n + \eta_2 y_n))}}, \\
y_{n+1} &= \frac{y_n}{y_n + (1-y_n)(1-\epsilon_2)^{N(\beta(\eta_1+\eta_2) - (\alpha+\beta)(\eta_1 x_n + \eta_2 y_n))}}.
\end{aligned}
\tag{46}
$$

After a similar change of variables as in the homogeneous case formula (46) becomes

$$
\begin{aligned}
x_{n+1} &= \frac{x_n}{x_n + (1-x_n)\exp(a_1(\eta_1 x_n + \eta_2 y_n - b))}, \\
y_{n+1} &= \frac{y_n}{y_n + (1-y_n)\exp(a_2(\eta_1 x_n + \eta_2 y_n - b))}.
\end{aligned}
\tag{47}
$$

In the simplest case of the equal shares/mixtures (i.e. $\eta_1 = \eta_2 = 0.5$) we have:

$$
\begin{aligned}
x_{n+1} &= \frac{x_n}{x_n + (1-x_n)\exp(a_1(0.5(x_n + y_n) - b))}, \\
y_{n+1} &= \frac{y_n}{y_n + (1-y_n)\exp(a_2(0.5(x_n + y_n) - b))}.
\end{aligned}
\tag{48}
$$

**Dimensional reduction**: Although the heterogeneous model contains more independent variables than the homogeneous case, the dynamics are constrained in a lower-dimensional manifold. That is, we will show that the function $I(x,y) = \frac{(1-x)^{a_2} y^{a_1}}{(1-y)^{a_1} x^{a_2}}$ is an invariant function for population mixtures. This means that the curves $I(x,y) = c$ are invariant for any time step $n$, where $c$ parametrizes the family of invariant curves.

**Lemma I.1.** *The function* $I(x,y) = \frac{(1-x)^{a_2} y^{a_1}}{(1-y)^{a_1} x^{a_2}}$ *is an invariant function (first integral) of the dynamics.*

*Proof.* It is easy to check that the set of equations (47) is equivalent to

$$
\begin{aligned}
\frac{x_{n+1}}{1 - x_{n+1}} &= \frac{x_n}{(1-x_n)\exp(a_1(\eta_1 x_n + \eta_2 y_n - b))}, \\
\frac{y_{n+1}}{1 - y_{n+1}} &= \frac{y_n}{(1-y_n)\exp(a_2(\eta_1 x_n + \eta_2 y_n - b))}.
\end{aligned}
\tag{49}
$$

By raising the first equation to power $a_2$ and the second equation to power $a_1$ and dividing them we derive that:

$$\frac{x_{n+1}^{a_2}(1-y_{n+1})^{a_1}}{(1-x_{n+1})^{a_2}y_{n+1}^{a_1}} = \frac{x_n^{a_2}(1-y_n)^{a_1}}{(1-x_n)^{a_2}y_n^{a_1}}$$

That is the function $I(x,y) = \frac{(1-x)^{a_2}y^{a_1}}{(1-y)^{a_1}x^{a_2}}$ is an invariant function (first integral) of the dynamics.

$\square$

**Time-average convergence of the mixture to Nash equilibrium** $b$. For the considered heterogeneous model we can show a result similar to Theorem 3.3 for the homogeneous population, that is that $b$ is Cesàro attracting mixture of trajectories.

**Theorem I.2.** *For every $a_1, a_2 > 0$, $b \in (0,1)$ and $(x_0, y_0) \in (0,1)^2$ we have*

$$\lim_{T \to \infty} \frac{1}{T} \sum_{n=0}^{T-1} (\eta_1 x_n + \eta_2 y_n) = b. \tag{50}$$

*Proof.* Let $f(x_n, y_n) = (x_{n+1}, y_{n+1})$ be defined by (47) where $\eta_1, \eta_2 \in (0,1)$ and $\eta_1 + \eta_2 = 1$.
The map $t \mapsto \frac{t}{1-t}$ is a homeomorphism of $(0,1)$ onto $(0,\infty)$, and its inverse is given by $t \mapsto \frac{t}{1+t}$.
Thus, we can introduce new variables, $z = \frac{x}{1-x}$ and $w = \frac{y}{1-y}$. In these variables our map will be
$g : (0,\infty)^2 \to (0,\infty)^2$, and if $g(z_n, w_n) = (z_{n+1}, w_{n+1})$, then

$$z_{n+1} = z_n \exp\left(-a_1\left(\eta_1\frac{z_n}{1+z_n} + \eta_2\frac{w_n}{1+w_n} - b\right)\right),$$
$$w_{n+1} = w_n \exp\left(-a_2\left(\eta_1\frac{z_n}{1+z_n} + \eta_2\frac{w_n}{1+w_n} - b\right)\right). \tag{51}$$

If $w_n = cz_n^{a_2/a_1}$ then $w_{n+1} = cz_{n+1}^{a_2/a_1}$. This shows that if $z_n$ is close to 0 then also $w_n$ is close to 0, and by (51) we get $z_{n+1} > z_n$. Similarly, if $z_n$ is close to infinity, then also $w_n$ is close to infinity, and by (51) we get $z_{n+1} < z_n$. Together with another inequality obtained from (51),

$$z_n \exp(-a_1(1-b)) < z_{n+1} < z_n \exp(a_1 b),$$

this proves that if $z_0, w_0 \in (0,\infty)$ then $\inf_{n\geq 0} z_n > 0$ and $\sup_{n\geq 0} z_n < \infty$.

The first equation of (51) can be rewritten as

$$z_{n+1} = z_n \exp(a_1(\eta_1 x_n + \eta_2 y_n - b)),$$

so by induction we get

$$z_T = z_0 \exp\left(a_1\left(\sum_{n=0}^{T-1}(\eta_1 x_n + \eta_2 y_n) - Tb\right)\right).$$

Therefore there exists a real constant $M$ (depending on the parameters and the initial point $(x_0, y_0)$), such that

$$\left|\sum_{n=0}^{T-1}(\eta_1 x_n + \eta_2 y_n) - Tb\right| \leq M$$

for every $T$. Dividing by $T$ and passing to the limit, we get

$$\lim_{T \to \infty} \frac{1}{T} \sum_{n=0}^{T-1} (\eta_1 x_n + \eta_2 y_n) = b. \tag{52}$$

$\square$

We end here with numerical results to demonstrate that, perhaps not surprisingly, this class of games not only possesses complex non-equilibrium behavior, but also allows for an immediate generalization to a more realistic, larger dimensional system, in which new and even more complex non-equilibrium phenomena can arise. Developing a more complete theoretical understanding of these issues, will likely require the introduction of new tools and techniques.

Figures (10) and (11) show attracting orbits generated from the map (48) (with $\eta_1 = \eta_2 = 0.5$) for fixed values of $a_1, a_2, b$. There, 5000 random starting points are initialized. To approximate where the attractors lie, the first 1000 iterates were made without plotting; the next 200 were visualized.

Figure 10: Attractor for the map (48) in the two-subpopulation model with $a_1 = 20, a_2 = 30, b = 0.8$. The white dots are the coordinates $(x, y)$ generated from initializing 5000 $(x_0, y_0)$'s at random from the unit square domain, iterating them with (48) 1000 times, then visualizing the next 200 iterates.

Figure 11: Attractor for the map (48) in the two-subpopulation model with $a_1 = 10, a_2 = 30, b = 0.7$. The white dots are the coordinates $(x, y)$ generated from initializing 5000 $(x_0, y_0)$'s at random from the unit square domain, iterating them with (48) 1000 times, then visualizing the next 200 iterates.

## J  Chaos in large atomic congestion games via reductions to the non-atomic case

In this paper we have focused on analyzing MWU in (mostly linear) non-atomic congestion games. In these settings each individual agent is assumed to control an infinitesimal account of the overall flow $N$. In the atomic setting each agent is controlling a discrete unsplittable amount of flow, i.e., a packet of size 1. There are now $N$ agents that need to choose amongst the different paths that are available to them. In this section, we will show how to translate results from the case of non-atomic congestion games to their atomic counterparts. To do so we will show that the MWU maps in the case of linear atomic congestion games can be reduced to MWU maps of non-atomic games, which we have already analyzed.

We will study MWU in a linear congestion game under the easiest information theoretic model of full information where on every day MWU receives as an input the expected cost of all actions/paths. Furthermore, we will assume that all $N$ agents are initialized with the fully mixed (interior) strategy $x$. This of course is not a generic initial condition but since we are working towards negative/complexity/chaos type of results we can choose our initial condition in an adversarial manner. Due to the symmetry of initial conditions, the payoff vectors that agents experience on any day are common across all agents. Hence, the symmetry of initial conditions is preserved. For such trajectories we only need to keep track of a single probability distribution (the same one for all agents), which is already reminiscent of the non-atomic setting where we only have to keep track of the ratios/split of the total demand along the different paths/strategies. Let's denote by $x$ this common probability vector for all $N$ agents. We are ready to define our model in detail.

**Atomic model with $N$ agents/players.** We have $N$ agents. Each agent can choose between $m$ strategies/paths. The cost function for each strategy/path is a linear function on the number of the agents using that path $i$, i.e., a linear function of its load. Let $\alpha_i$ be the respective multiplicative constant for strategy $i$. Suppose all agents use the same probability distribution $x$. The expected cost of any agent for using strategy $i$ is

$$c(i) = \alpha_i(1 + (N-1)x_i) \ \forall i \in \{1, \ldots, m\}. \tag{53}$$

Given this payoff vector, the MWU updates follow the same format as always. At time $n+1$ the players know already the expected cost of the strategies at time $n$ and update their choices. The update rule in the case of $m$ strategies is as follows:

$$x_i(n+1) = x_i(n) \frac{(1-\epsilon)^{c(i)}}{\sum_{j \in \{1,\ldots,m\}} x_j(n)(1-\epsilon)^{c(j)}}, \tag{54}$$

We are now ready to state two formal results. One for the case of games with two strategies and one for the more general case with $m$ strategies.

**Theorem J.1.** *Let's consider an atomic congestion game with $N$ agents and two paths of linear cost functions as described by equations (53), (54). Let $x$ be an interior probability distribution that is a common initial condition for all $N$ agents. As long as the congestion game has a symmetric interior Nash equilibrium where both agents play the distribution $(p, 1-p)$ with $0 < p < 1$[13] the update rule of the probability distribution $x$ under MWU dynamics is as in the case of the non-atomic model map (4) where $a = (N-1)(\alpha_1 + \alpha_2)\ln\left(\frac{1}{1-\epsilon}\right)$ and $b = p$. Thus, as long as $p \neq 0.5$, there exists a threshold capacity $N_0$ such that if the number of agents $N$ exceeds $N_0$ the system has periodic orbits of all possible periods, positive topological entropy and is Li-Yorke chaotic. If $p = 0.5$, although the Price of Anarchy of the game converges to one as $N \to \infty$, the time-average social cost can be arbitrarily close to its worst possible value.*

*Proof.* By substituting into (54) the values of the cost functions from (53) we get:

$$
\begin{aligned}
x_{n+1} &= \frac{x_n(1-\epsilon)^{\alpha_1(1+(N-1)x_n)}}{x_n(1-\epsilon)^{\alpha_1(1+(N-1)x_n)} + (1-x_n)(1-\epsilon)^{\alpha_2(1+(N-1)(1-x_n))}} \\
&= \frac{x_n}{x_n + (1-x_n)(1-\epsilon)^{\alpha_2 N - \alpha_1 - (\alpha_1 + \alpha_2)(N-1)x_n}}.
\end{aligned}
\tag{55}
$$

We introduce the new variables

$$a = (N-1)(\alpha_1 + \alpha_2)\ln\left(\frac{1}{1-\epsilon}\right), \quad b = \frac{\alpha_2 N - \alpha_1}{(\alpha_1 + \alpha_2)(N-1)}. \tag{56}$$

Note that the symmetric strategy where all agents play according to $(b, 1-b)$ is an interior Nash equilibrium. Given this new formulation we see that the map is the same as the one for the non-atomic case (4). The claims about chaos follow by direct application of Corollary 3.10. In the case where $p = 0.5$, we have that the uniform distribution is an interior Nash and this implies that $\alpha_1 = \alpha_2(= \alpha)$,

i.e. both paths have the same cost function. In terms of Price of Anarchy, the expected cost of any agent at a Nash equilibrium is at most $\alpha(1 + \frac{N-1}{2}) = \alpha\frac{N+1}{2}$. Hence the social cost of any Nash equilibrium is at most $\alpha\frac{N(N+1)}{2}$. On the other hand, the socially optimal state that divides the load as equally as possible has cost at least $\alpha\frac{N^2}{2}$ and the ratio of the two converges to 1 as $N \to \infty$. Finally, the fact that there exist trajectories such that the time-average social cost can be arbitrarily close to its worst possible value follows from a direct application of Theorem 5.1 given the equivalence of the update rule for the atomic and non-atomic case and the fact for both systems (approximate) worst case performance is experienced when (in expectation almost) all users/flow are using the same strategy. $\qquad\square$

Theorem J.1 applies for atomic congestion games with numerous agents but only two paths. As we show next, chaos is robust and emerges in atomic congestion games regardless of the number of available paths.

**Theorem J.2.** *Let's consider an atomic congestion game with $N$ agents and $m$ paths of linear cost functions as described by equations (53), (54). Let the cost functions of the all paths be $\alpha x$ where $x$ the load of the respective path and $\alpha$ the common multiplicative constant. Let $x$ be interior probability distribution that is a common initial condition for all $N$ agents. The update rule of the probability distribution $x$ under MWU dynamics is as in the case of the non-atomic model map (28) where $a_i = (N-1)\alpha\ln\left(\frac{1}{1-\epsilon}\right)$. Thus, for any such atomic congestion game there exists a threshold capacity $N_0$ such that if the number of agents $N$ exceeds $N_0$ the system has periodic orbits of all possible periods, positive topological entropy and is Li-Yorke chaotic.*

*Proof.* The reduction of the map described by equations (53), (54) to non-atomic model map (28) follows easily once we observe that MWU, i.e. map (54) is invariant to shifts of the cost vector by a constant value.[14] That is, for any $\gamma$ if we apply the vector $c'(i) = c(i) + \gamma$ to map (54) it remains unchanged. Hence instead of substituting into (54) the values of the cost functions $c(i) = \alpha(1 + (N-1)x)$, we instead substitute the values $c'(i) = \alpha(N-1)x$. However, this is exactly map in the case of the non-atomic model map (28) with $a_i = (N-1)\alpha\ln\left(\frac{1}{1-\epsilon}\right)$. The rest of the theorem follows immediately by applying Theorem G.1. $\qquad\square$

# K  Other related work

**Main precursors.** [42] put forward the study of chaotic dynamics arising from Multiplicative Weights Update (MWU) learning in congestion games. They established the existence of an attracting limit cycle of period two and of Li-Yorke chaos for MWU dynamics in *atomic* congestion games with two agents and two links with linear cost functions. Symmetry of the game (i.e., the existence of a symmetric equilibrium where both agents select each path with probability 0.5) results in a limit cycle of period two. They also showed for a *specific instance* of a game with an asymmetric equilibrium that MWU leads to Li-Yorke chaos, provided that agents adapt the strategies with a sufficiently large learning rate (step size) $\epsilon$ (equivalently, if agents use a fixed learning rate $\epsilon$ but their costs are scaled up sufficiently large). Shortly afterwards, [14] established that Li-Yorke chaos is prevalent in *any* two-agent *atomic* congestion games with two parallel links and linear cost functions, provided the equilibrium is asymmetric. Namely, in *any* $2 \times 2$ congestion game with an asymmetric equilibrium, Li-Yorke chaos emerges as the cost functions grow sufficiently large, but only if the initial condition is symmetric, i.e., both agents start with the same initial conditions. Furthermore, [14] established for the first time that, despite periodic or chaotic behaviors, the time-average strategies of both agents *always* converge *exactly* to the interior Nash equilibrium. While our current work leverages techniques from [14], it also investigates other definitions of chaos, e.g., positive topological entropy, studies *non-atomic* congestion games, and relates the results to the Price of Anarchy and system efficiency analysis. Moreover, whereas in [14, 42] chaotic behavior is contained in a one-dimensional invariant subspace of the two dimensional space, in this paper the dimensionality of the system is already equal to one and hence the chaotic results are relevant for the whole state space. Lastly, in the appendices, we provide preliminary results for learning dynamics in larger and more complex congestion games with many degrees of freedom.

**Chaos in game theory.** Under the assumption of perfect rationality, it is not surprising that Nash equilibria are central concepts in game theory. However, in reality, players do not typically play a game following a Nash equilibrium strategy. The seminal work of [49] showed analytically by computing the Lyapunov exponents of the system that even in a simple two-player game of rock-paper-scissor, replicator dynamics (the continuous-time analogue of MWU) can lead to chaos, rendering the equilibrium strategy inaccessible. For two-player games with a large number of available strategies (complicated games), [25] argue that Experienced Weighted Attraction (EWA) learning, a behavioral economics model of learning dynamics, exhibits also chaotic behaviors in a large parameter space. The prevalence of these chaotic dynamics also persists in games with many players, as shown in the recent follow-up work [48]. Thus, careful examinations suggest a complex behavioral landscape in many games (small or large) for which no single theoretical framework currently applies. [53] and [56] prove that fictitious play learning dynamics for a class of 3x3 games, including the Shapley's game and zero-sum dynamics, possess rich periodic and chaotic behavior. [13] prove that many online learning algorithms, including MWU, with a constant step size is Lyapunov chaotic when applied to zero-sum games. Finally, [43] has established experimentally that a variant of reinforcement learning, Experience Weighted Attraction, leads to limit cycles and high-dimensional chaos in two agent games with negatively correlated payoffs. This is strongly suggestive that chaotic, non-equilibrium results can be further generalized for other games.

## Footnotes

[7]Therefore $b \in (x_l, x_r)$ when $x = b$ is repelling (for $a > 2/b(1 - b)$).

[8]Recall from (3) that $a = N \ln\left(\frac{1}{1-\epsilon}\right)$.

[9]In this way, we can numerically approximate the *signed* second Feigenbaum constant $\alpha$ [54].

[10]Recall that the orbit is superstable if one of the critical points is an element of the orbits, so that $f'_{a,b}(x_c) = 0$. This means the Lyapunov exponents in principle is $-\infty$, visualized as a white bright color.

[11]Given that the case $m=2$ with $\alpha_1 = \alpha_2$ is analyzed in Theorem 3.7 (emergence of a periodic orbit of period 2) we have a complete understanding of all cases.

[12]This is convenient as it immediately implies that the Price of Anarchy is equal to 1, since the potential is equal $\frac{1}{p+1}$ of the social cost function in these games and thus the equilibrium flow minimizes both the potential and the social cost.

[13]The game has a symmetric interior Nash if and only if $\alpha_2 < N\alpha_1$ and $\alpha_1 < N\alpha_2$.

[14]This invariance is also true for most standard regret minimizing dynamics, e.g. Follow-the-Regularized-Leader.