[Reviews · NeurIPS 2020]

Review 1

Summary and Contributions: This paper studies learning dynamics in games. Agents are not perfectly rational, but rather use multiplicative-weight updating (MWU) to select their actions over time. The focus is on non-atomic routing games in simple networks. The authors explore how the dynamics changes as the system load changes (i.e., as the total volume of traffic increases). The high-level conclusion is that while the time-averaged behavior approaches a Nash equilibrium, the dynamics themselves become more chaotic and less likely to converge to a stable solution as load increases. The intuition is that higher load leads to higher-magnitude costs, which increases the sensitivity of MWU beyond the point where convergence is guaranteed. Previous work had argued that the price of anarchy in routing games approaches 1 as load increases. This work makes the counter-point that higher load can make it less likely for the system to reach equilibrium. The techniques are a combination of theoretical analysis and empirical simulation.

Strengths: I really like the conceptual point being made in this paper. Analysis of off-equilibrium dynamics is a tough nut to crack, and the authors make an interesting observation about the effect of cost changes on learning. The paper is well-written. Strong technical contribution, interesting conceptual point.

Weaknesses: Specific to a particular learning model. Reliant on assumptions (constant learning rate, unnormalized payoffs). I elaborate more on these points below.

Correctness: As far as I understand the claims and methodology are correct.

Clarity: The paper is well-written.

Relation to Prior Work: The relationship with prior work is clear.

Reproducibility: Yes

Additional Feedback: The results rely crucially on modeling assumptions. In particular, it is assumed that all agents are using the same (standard) MWU dynamics, and that they all use the same learning parameter epsilon. I found myself wondering whether the existence of the chaotic regime would still hold if agents used different values of epsilon, say distributed continuously (without atoms) over the population. Even if it weren't possible to prove results, performing some robustness checks of that form using empirical simulations would help make the point that this is a robust phenomenon. When making comparisons across loads, it's important for the analysis that the agents do not scale their payoffs to lie in [0,1]. I'm on the fence about the motivation for this. If an agent is repeatedly interacting with a system at a fixed load, I'd expect them to adapt their learning to that environment, which (arguably) corresponds to scaling to [0,1]. On the other hand, if we're thinking about a setting where the load of a system is actually changing (slowly) over time, I think it's reasonable that agents wouldn't update their scaling in response. E.g., they express "undue frustration" if they suddenly start experiencing large delays where they weren't before. So I think this modeling choice is somewhat dependent on the application. Post-rebuttal: the authors helped lift some of the worries I had about robustness to multiple learning rates and the motivation behind their form of scaling. I now lean more positively about the paper, and I have adjusted my score accordingly.


Review 2

Summary and Contributions: This paper studies the dynamics of multiplicative weights updates (MWU) in routing games. Their analysis focuses on the case where cost is not normalized to be in the range [0,1]. This is outside the operational regime MWU is typically studied in, but is well motivated by work in behavioral economics and serves as a stress test for MWU. The analysis in the main body of the paper focuses on routing games with non-atomic agents, linear costs, and two strategies. In this setting the authors derive a 1-dimensional iterative map of the unit interval. By studying this interval map, they prove that the time average of the map converges to the equilibrium since the map has orbits with center of mass equal to the equilibrium. They also show that by varying parameters it can lead to chaotic behaviours (specifically, by increasing the total demand in the system when the equilibrium isn’t symmetric). Finally they show that time average regret is bounded and that the time average social cost can be arbitrarily close to the worst case. In the supplementary materials the authors generalize their results in numerous ways (e.g. multi-strategy settings) and provide extensive numerical examples.

Strengths: 1. Interesting problem that deviates in a natural way from typical questions asked in the literature. 2. Results provide an important cautionary tale for any learning applications of routing games. Also, has an interesting message that applies beyond just routing games; it opens up the question of whether similar phenomena might arise in other game theoretic learning settings with similar information structure and "unbalanced" signaling/incentives (e.g. markets). 3. Applies techniques from dynamical systems theory to game theoretic learning in novel ways. These proof techniques may be applicable to other settings and demonstrate the kinds of new rigorous statements that can be proved with these techniques. 4. When considered alongside the supplementary materials, I felt that the paper leaves very few questions about MWU in the setting they studied. 5. The authors did a good job of explaining how their results relate to the rest of the literature. I also think they did a good job of motivating their setting in the Introduction. 6. Proofs in the supplementary materials were well written.

Weaknesses: 1. Assumes background and intuition about dynamical systems theory, which a general audience at Neurips may not be familiar with. Note, some of this is provided in the supplementary materials. 2. Clarity seems to have been sacrificed to meet the page limit. Important sections of the paper felt terse, unnecessarily dense, and lacking intuition (e.g. section 4). Also, I felt that many of the results that elevate the appeal of this paper are not highlighted in the way they should be since they are in the supplementary materials (e.g. the relation to atomic routing games and the generalization to multiple strategies). 3. Neurips may not be the most appropriate setting. I personally believe it is within the scope, but rather tangentially. -------------- Post-Rebuttal/Discussion: My points above have been partially addressed, but I am generally less concerned about them. In particular, I'm still a little worried that an audience that is less familiar with these techniques will get very little out of anything but the introduction without having to read the supplementary materials. That being said, given the updated information/discussion, and that I really like the ideas and techniques in the paper, I'll be updating my score.

Correctness: Yes. The claims are rigorously stated in the main body of the paper and the proofs in the supplementary materials are well written.

Clarity: Certain sections are very well written (particularly the Introduction). However, many of the technical portions of the paper seemed highly condensed and lacked intuition. I would like to note that no part of the paper was poorly written per se, but the reader may have a hard time parsing this paper unless the reader is well acquainted with dynamical systems and routing games.

Relation to Prior Work: Yes. I thought they did quite a good job of motivating their work's relation with the rest of the literature.

Reproducibility: Yes

Additional Feedback: Suggestions for the Authors: 1. The second paragraph of the Abstract is rather verbose in its explanation of the symmetric flow case. I'd suggest condensing this into one or two sentences at most. 2. The paragraph in the Introduction starting on line 48 doesn't address problems of the cost normalization assumption from a behavioral economics perspective. A sentence or two explaining this would make the paragraph much more impactful and motivate your setting better. 3. In section 2 it may help the reader understand the parameters a and b (the normalized system demand and equilibrium flows) if you had presented a little more intuition about the parameters \alpha and \beta when originally defining the cost functions. 4. It may be worthwhile to explicitly state early in section 2 that you will assume \alpha + \beta = 1 throughout the paper. You state this numerous times and it would save you space. 5. All links in the PDF to Figures other than 1 point to the header of the document. You use these to support certain claims, so without being able to see the figures these statements seem unjustified and the reader must verify these claims on their own (e.g. the statement about fixed points of the interval map on line 209). Unless the reader has the supplementary materials readily available, this could be confusing and might warrant a verbal explanation. 6. Adding some more explanation about certain claims could greatly improve the exposition with minimal additional verbiage required. For example, on line 211, a brief reminder for the reader of exactly what about the derivatives at the fixed points makes them repelling would make it easier to read without having to possibly check back on definitions. Similarly, on line 212, the same principle applies to the second degree polynomial you present. Small examples like these are throughout the paper, but I imagine this is in part to meet the page restriction. Typos and Misc. Writing Comments: Abstract) - Lines 8-9. Some confusing wording in the last sentence of the first paragraph (in the sentence describing the benefits of not normalizing costs). Introduction) - Line 47. "...approximately optimality." --> "...approximate optimality." - Line 51. "These assumptions although standard in the online optimization literature they are far from the norm..." --> "These assumptions, although standard in the online optimization literature, are far from the norm..." - Line 71. "As a first step in direction of understanding the effects of increased demand..." --> "As a first step in the direction of understanding the effects of increased demand..." - Line 94-95. Upon first reading this seems to contradict the previous sentence's claim that time-average costs converge to equilibrium in the linear cost setting. You should probably be a little more careful to clarify what is meant here. Section 4) - Line 286-288. I believe there is an extra "Lebesgue" mentioned in this sentence. Conclusion) - Line 313. I believe you meant "Exploring further upon these network of connections for different dynamics, games is a fascinating..." --> "Exploring this network of connections further for different dynamics and games is a fascinating..."


Review 3

Summary and Contributions: This paper considers the dynamics of the unnormalized MWU when applied for non-atomic two-strategy congestion games with linear cost functions. The chaotic features is characterized theoretically, with experimentally illustration shown in supplementary materials. Although existing literature has proved the convergence properties of MWU on congestion games under certain normalization assumptions, this paper shows the dynamics of MWU can be wildly chaotic with bifurcation, periodic orbits emerging, as the demand of the roads/strategies increases and exceeds a certain threshold.

Strengths: The idea and results in this paper are novel. Precisely, it reveals the chaotic dynamic features of MWU on congestion games as demand increases, which is counter-intuitive to the known results about MWU's low PoA and the convergence of its time average to Nash equilibrium. The technical analysis for the dynamics are well-writen and relatively complete.

Weaknesses: How the choatic phenomenons influence the application of MWU in practice is not discussed. The studied games are relatively simple, even those games which the author claims the results can be extended to. The formula for the minimal regret (the equation after Eqn. (6)) seems to be wrong. (\min \left\{\sum_{n=1}^{T} \alpha N x_{n}, \sum_{n=1}^{T} \beta N\left(1-x_{n}\right)\right\} instead of \min \left\{\sum_{n=1}^{T} \alpha N, \sum_{n=1}^{T} \beta N\right\})

Correctness: I believe the claims in the main content are correct, but did not check all the proofs in Appendix.

Clarity: The presentaion of this paper can be improved. Commas are missing in some places. Many figures in supplementary materials are referred directly in main content. The text in main content should be self-contained.

Relation to Prior Work: It is clearly discussed how this work differs from previous contributions.

Reproducibility: Yes

Additional Feedback:

[Author Response · NeurIPS 2020]

We would like to thank the reviewers for their interesting comments and questions.

**Reviewer 1:** It is great to hear that you really like the conceptual point of our paper and that you appreciate our strong
technical contribution.

**Q1:** *Does chaotic behavior persist if agents use different values of epsilon?* Great question! In Appendix I, pages
33-36 of the full paper, we examine the case of two learning rates $\epsilon_1$ and $\epsilon_2$ with both theoretical and experimental
results (see Figs 11, 12). These results can be extended to any finite number of learning rates by paying mostly a
notational overhead. Rev. 1 has an excellent idea about a continuum of learning rates distributed (without atoms) over
the population. This is a very interesting open question. Novel ideas and techniques would be required. Unfortunately,
this setting is not really conducive to simulations as the full system state is a continuous measure. A discretized version
with a finite large set of learning rates would result in a more complex version of our $\epsilon_1$ and $\epsilon_2$ figures. We will be
happy to expand upon this in a discussion section for future work.

**Q2:** *Scaling of costs in $[0, 1]$:"If an agent is repeatedly interacting with a system at a fixed load, I'd expect them to*
*adapt their learning to that environment, which (arguably) corresponds to scaling to $[0, 1]$."* There is a subtle but
important distinction to be made here. The normalization you describe here does not result in games with costs in $[0, 1]$,
not even approximately. Let's assume when you travel to work it takes somewhere between 10 to 30 minutes and hence
you normalize your costs by dividing by 30 or even 60 to be safe. Your worst case cost can still be arbitrarily larger
than 1 especially in games with large populations as it corresponds to the case where you choose the longest possible
route and *everyone else* in your city chooses that route too. If one was to work on a formal model of the suggested
normalization two things should be clear: 1) whether chaos emerges or not still has to be carefully explored using our
techniques 2) this is not the normalization in PoA literature that assumes that not just *your typical operating costs* but
the much more demanding condition that even *worst case costs* lie in $[0, 1]$. *" if . . . the load of a system is actually*
*changing (slowly) over time, I think it's reasonable that agents wouldn't update their scaling in response."* We agree
and we want to point out that in (Lykouris et al. Learning and efficiency in games with dynamic population, SODA
2016) although the authors explicitly focus on a time-dependent population size, they assume that *all* stage game (worst
case) costs are bounded in [0,1] (e.g. Theorem 3.1). Since this modelling assumption is not always easily applicable
and as we show it can totally dictate system performance, we need to study it carefully.

**Reviewer 4:** Thank you for enumerating such a long list of strengths for our paper.

In regards to the background on dynamical systems, as you point out, we included materials both in the main part
as well as the full paper. Although the theoretical analysis is technical, the message itself is easily understood even
by non-experts. We believe that there is important value and insights both for theoretical minded people as well as
experimentalists, especially given our very thorough experimental study with numerous figures. We agree that NeurIPS
space constraints are tight and have forced us to push the presentation of some important results, as you point out, into
the appendix. Nevertheless, all of our results are at least described completely in the introduction and we hope that the
interested reader would benefit from our expansive supplement and directly jump in the section that they find more
interesting. Finally, we respectfully disagree that our paper is only tangentially related to NeurIPS. The Palaiopanos et
al. paper was published at NeurIPS in 2017 with a spotlight distinction. Since then it has received 46 citations including
from at least 8 other NeurIPS papers. The ideas relating to period 3 inducing Li-Yorke chaos in dynamical systems have
since then found new applications related to the representational power of Deep Neural Networks: Chatziafratis et al.
Depth-Width Trade-offs for ReLU Networks via Sharkovsky's Theorem ICLR '20 (spotlight); Chatziafratis et al. Better
Depth-Width Trade-offs for Neural Networks through the lens of Dynamical Systems ICML '20.

We are very grateful for your detailed comments. We will incorporate them in an updated version of our paper.

**Reviewer 7:** Thank you for your support of our paper.

*Are there implications for MWU in practice?* In applications Exp3, the bandit version of MWU, is typically used.
Recent work on wireless network selection, which is a congestion game, has shown that Exp3 is unstable and has bad
performance necessitating new domain-specialized algs (Oh et al. Periodic Bandits and Wireless Network Selection
ICALP '19, Appavoo et al. Shrewd Selection Speeds Surfing: Use Smart EXP3! Int. Conf. on Distributed Computing
Systems '18). These issues are important and largely unsolved from a practical perspective.

*The studied games are relatively simple, even those games which the author claims the results can be extended to.* The
emergence of chaos is clearly a hardness/complexity type of result. Such results only increase in strength the simpler
the class of examples is. Even a single instance of a bad example suffices as, e.g., in Palaiopanos et al. We instead, show
that *all* simple networks, no matter their costs, number of links, (except for the case of two links with equal costs) will
*all* exhibit chaos, *every single instance of them*. The simplicity, robustness of our games is a major strength of our paper.

Our regret formula is correct as $\min\{\sum_{n=1}^{T} \alpha N x_n, \sum_{n=1}^{T} \beta N (1 - x_n)\}$ captures the minimum aggregate cost over
time of the two paths. We need the knowledge of $x_n$ to compute that.

[Meta-Review · NeurIPS 2020]

The reviewers were generally positive about the paper, and the rebuttal addressed all major concerns that remained; we are happy to recommend acceptance.